# Perceptual saccadic suppression starts in the retina

Saad Idrees 1,5, Matthias P. Baumann1,2,5, Felix Franke3, Thomas A. Münch1,4✉ & Ziad M. Hafed 1,2✉

Visual sensitivity, probed through perceptual detectability of very brief visual stimuli, is strongly impaired around the time of rapid eye movements. This robust perceptual phenomenon, called saccadic suppression, is frequently attributed to active suppressive signals that are directly derived from eye movement commands. Here we show instead that visual-only mechanisms, activated by saccade-induced image shifts, can account for all perceptual properties of saccadic suppression that we have investigated. Such mechanisms start at, but are not necessarily exclusive to, the very first stage of visual processing in the brain, the retina. Critically, neural suppression originating in the retina outlasts perceptual suppression around the time of saccades, suggesting that extra-retinal movement-related signals, rather than causing suppression, may instead act to shorten it. Our results demonstrate a far-reaching contribution of visual processing mechanisms to perceptual saccadic suppression, starting in the retina, without the need to invoke explicit motor-based suppression commands.

[1] Werner Reichardt Centre for Integrative Neuroscience, Tübingen University, 72076 Tübingen, Germany. [2] Hertie Institute for Clinical Brain Research, Tübingen University, 72076 Tübingen, Germany. [3] Bio Engineering Laboratory, ETH Zürich, 4058 Basel, Switzerland. [4] Institute for Ophthalmic Research, Tübingen University, 72076 Tübingen, Germany. [5] These authors contributed equally: Saad Idrees, Matthias P. Baumann. ✉email: thomas.muench@uni-tuebingen.de; ziad.m.hafed@cin.uni-tuebingen.de

S accadic eye movements are a prominent feature of visual behavior; they allow successive sampling of information from the environment. However, from the perspective of visual information flow into the brain, these rapid eye movements constitute highly disruptive events, introducing spurious motions that should normally go perceptually unnoticed, or canceled. The question of how and why such perceptual cancelation takes place has intrigued philosophers and scientists for many decades[1–4]. Indeed, sensitivity to brief peri-saccadic visual probes is strongly impaired, in a phenomenon known as saccadic suppression that has repeatedly been demonstrated in a multitude of experiments[5–14].

Despite the robustness of saccadic suppression as a perceptual phenomenon, the mechanisms behind it remain highly controversial. On the one hand, perceptual suppression may arise through internal knowledge of planned eye movements and their associated motor commands[5,12,15–18]. According to this popular view, eye movement commands are a necessary prerequisite for saccadic suppression: a movement-related signal[16,17], such as corollary discharge from (pre-)motor areas, may act as a suppressive command for visual neurons to cause perceptual suppression, and maybe even in a pathway-selective manner[10].

On the other hand, perceptual saccadic suppression could also arise as a result of the visual consequences of retinal image shifts[2,19–30]. After all, the early visual system, including the retina, is a highly sensitive light sensing device, and can capture visual transients associated with saccade-induced retinal image shifts. Such early processing of visual transients could modulate the retinal output, jumpstarting an image processing cascade to mediate perceptual suppression.

In this study, rather than arguing either strictly for or against one of these seemingly contrasting hypotheses, we asked to what extent they might interact with and support each other to ultimately serve perception. We were specifically motivated by the fact that the very first visual processing stage in the brain, the retina, is not only sensitive to visual transients (such as saccade-induced image shifts), but it also possesses rich image processing circuitry that could regularize the visual disruptions[31–35] caused by saccades. We therefore asked: how much of the characteristics of perceptual saccadic suppression can be explained by visual-only mechanisms? And, to the extent that there are visual-only mechanisms, would the first neural locus for them indeed be the very first stage of visual processing in the brain, the retina?

We used a multi-disciplinary approach in which we experimentally mimicked the visual consequences of saccades and recorded neural activity from ex vivo retinae of different animal models. We also measured human perceptual reports using both real saccades and saccade-like image displacements to simulate the saccadic visual flow. We found a surprisingly far-reaching contribution of visual processing mechanisms to perceptual saccadic suppression, starting in the retina, without the need to invoke explicit motor-based suppression commands. Intriguingly, the role of motor-based commands seems to be the opposite of what has been proposed before. Rather than sending an explicit suppressive command to reduce visual system sensitivity, motor-based commands instead seem to minimize the duration of visually derived saccadic suppression.

## Results

**Perceptual saccadic suppression depends on image content**. We first asked human subjects to generate saccades across textured backgrounds, akin to how saccades may be made in real life. Subjects viewed coarse or fine textures (Fig. 1a, Methods and Supplementary Fig. 1). Starting from one of four locations on the display, subjects made 4.8 deg saccades towards display center

(Fig. 1a, left). We varied saccade onset and endpoint locations, as well as texture images, across trials to avoid subjects remembering specific texture patterns. At a random time, a luminance pedestal (probe flash) was added to the texture background, for one display frame (~12 ms), at one of four locations relative to saccade endpoint (7 deg eccentricity; Fig. 1a, right). Subjects localized the probe flash (4-alternative-forced-choice paradigm), and we analyzed how well they did so. We ensured that the retinal region of flash location was stimulated with the background texture (rather than the edge of the monitor or the black surround of the dark laboratory) throughout any given trial, and that the probe flash was larger than the image blobs in the coarse texture, such that average luminance variation within each flash was matched across trials and textures. Coarse and fine textures had blobs that approximated the sizes of retinal ganglion cell (RGC) or retinal bipolar cell receptive fields, respectively, at the retinal flash locations[36] (Methods).

For both coarse and fine textures, subjects were strongly impaired in their ability to localize peri-saccadic flashes, thus experiencing strong perceptual saccadic suppression (Fig. 1b, c). Importantly, the suppression clearly depended on background visual images: it started earlier and recovered later with saccades across coarse rather than fine textures (Fig. 1d; the highlighted time intervals show significant differences between coarse and fine textures with $p < 0.001$, cluster-based random permutation test[37,38]). Moreover, the peak amount of suppression was stronger with the coarse textures (Fig. 1d). However, for both types of textures, performance reached a floor effect, masking an even larger difference (addressed below and in Fig. 2). This dependence of perceptual saccadic suppression on background texture was robust across individual subjects (Supplementary Fig. 2a; also see Supplementary Fig. 4 for further individual subject effects).

To rule out the possibility that flashes might simply be easier to see over the fine texture, we performed a control experiment in which we collected full psychometric curves of perceptual performance during fixation. Without any saccades, probe flash visibility was identical over coarse and fine textures (Supplementary Fig. 3a, b). Therefore, the image dependence demonstrated in Fig. 1 was related to saccadic suppression itself and not to the baseline visibility of brief flashes over the different textures. Similarly, analyzing eye movement properties showed that the results of Fig. 1 were also not due to different saccade kinematics for the different textures (Supplementary Fig. 3c, d).

We next employed a more sensitive procedure to evaluate perceptual thresholds. We repeated the same experiment of Fig. 1 on five subjects (three being the same as in the earlier experiment). This time, however, we collected full psychometric curves (Methods; similar to Supplementary Fig. 3a, b). As collecting full psychometric curves for each texture and each time point relative to saccade onset would be a very data-intensive endeavor, we expedited data collection by implementing a real-time saccade detection algorithm, described by Chen and Hafed[39]. This allowed us to present the probe flash at only four defined times after online saccade detection, strategically chosen to evaluate peak suppression (shortly after saccade onset), as well as the recovery time course after a saccade. We used an adaptive QUEST[40] procedure to estimate perceptual threshold per condition and flash time (Methods), with perceptual threshold (for the purposes of QUEST) being defined as the flash contrast value resulting in 62.5% correct performance. Besides the QUEST procedure, we also collected more trials showing different flash contrast levels relative to the estimated threshold, in order to obtain full psychometric curves. The results are shown in Fig. 2, and they match those of Fig. 1: relative to the baseline psychometric curves of flash visibility long after saccades (dashed

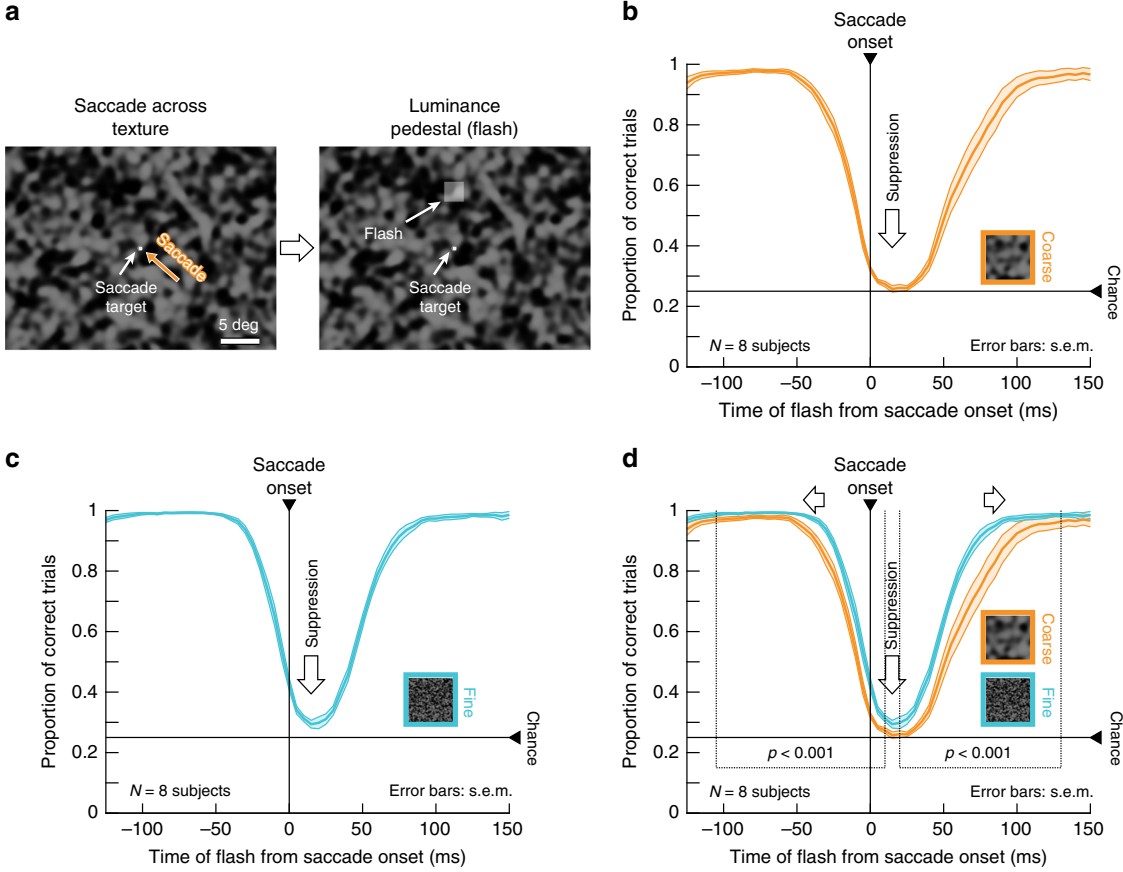

**Fig. 1 Image dependence of perceptual saccadic suppression. a** Human subjects generated saccades across a texture (here: coarse) from one of four diagonal locations towards display center (here: from the lower right). A luminance pedestal was flashed peri-saccadically at one of four locations around display center (right, left, up, or down; here: up). The insets in **c**, **d** show fine textures for comparison; also see Supplementary Fig. 1 and Methods. **b**, **c** Subjects failed to localize peri-saccadic flashes with both coarse (**b**) and fine (**c**) textures (perceptual reports were binned as a function of flash time from saccade onset using 50-ms bins moving in steps of 5 ms). **d** Perceptual suppression started earlier and lasted longer with a coarse background (also see Fig. 2). The highlighted times denote significantly different (p < 0.001, two-tailed random permutation test) time clusters between coarse and fine conditions (Methods). Curves show averages ± s.e.m. of individual subjects' suppression curves (N = 8). Supplementary Figs. 2, 3 show individual subject results, as well as controls for flash visibility (in the absence of saccades) and saccade motor variability.

curves), peri-saccadic psychometric curves were clearly shifted towards higher thresholds (Fig. 2a–d), consistent with Fig. 1. Critically, the more sensitive approach of full psychometric curves revealed that perceptual saccadic suppression was much stronger for coarse than fine textures at peak suppression; that is, perceptual thresholds (defined as luminance increments required for a specific correct performance level; Methods) near peak suppression were higher for coarse than fine textures (Fig. 2e). Supplementary Fig. 4 shows the individual subject psychometric curves.

To summarize, perceptual saccadic suppression is associated with a visual component directly influencing its strength and time course: saccades across coarse textures are associated with both stronger and longer-lasting perceptual suppression than saccades across fine textures, even when eye movement kinematics (and thus underlying motor commands) are controlled for.

**Perceptual saccadic suppression originates in the retina**. To test if this visual component of perceptual saccadic suppression originates in the retina, we isolated mouse and pig retinae and performed multi-electrode array recordings (Methods). We continuously exposed each retina to coarse and fine textures, matched to ganglion and bipolar cell receptive field sizes in the recorded species (Supplementary Fig. 1). We rapidly translated

the textures to simulate saccade-like image displacements (Fig. 3a). Such displacements can robustly activate RGCs, as is evident from the example mouse RGC shown in Fig. 3b. In fact, most recorded RGCs (mouse: 83% of 1,423 cells, pig: 73% of 394 cells) responded to texture displacements, indicating that saccade-induced visual transients during active gaze behavior constitute strong signals to the retina. Next, at different times relative to texture displacements, we introduced a luminance pedestal (probe flash) to the entire texture for 16 or 33 ms, similar in principle to the perceptual experiments of Figs. 1 and 2. Such flashes, when presented in isolation (that is, temporally removed from texture displacements), elicited responses in a sizable fraction of RGCs (baseline response; mouse: 688 of 1423 RGCs; pig: 228 of 394 RGCs). This allowed us to evaluate the consequences of texture displacements on flash responses in these cells—conceptually similar to the experiments in Figs. 1 and 2 (in which we evaluated the consequences of saccades on flash perception). The same example RGC of Fig. 3b showed much suppressed neural responses to the flash when it was presented immediately after texture displacements compared to baseline (Fig. 3c, d). This suppression of flash-induced responses (Fig. 3d) looks remarkably similar to suppression of visual responses in, say, macaque superior colliculus for stimuli presented after real saccades[13,14,41]. Thus, neuronally, there does exist "saccadic suppression" of visual sensitivity at the very first stage of visual processing, the retina,

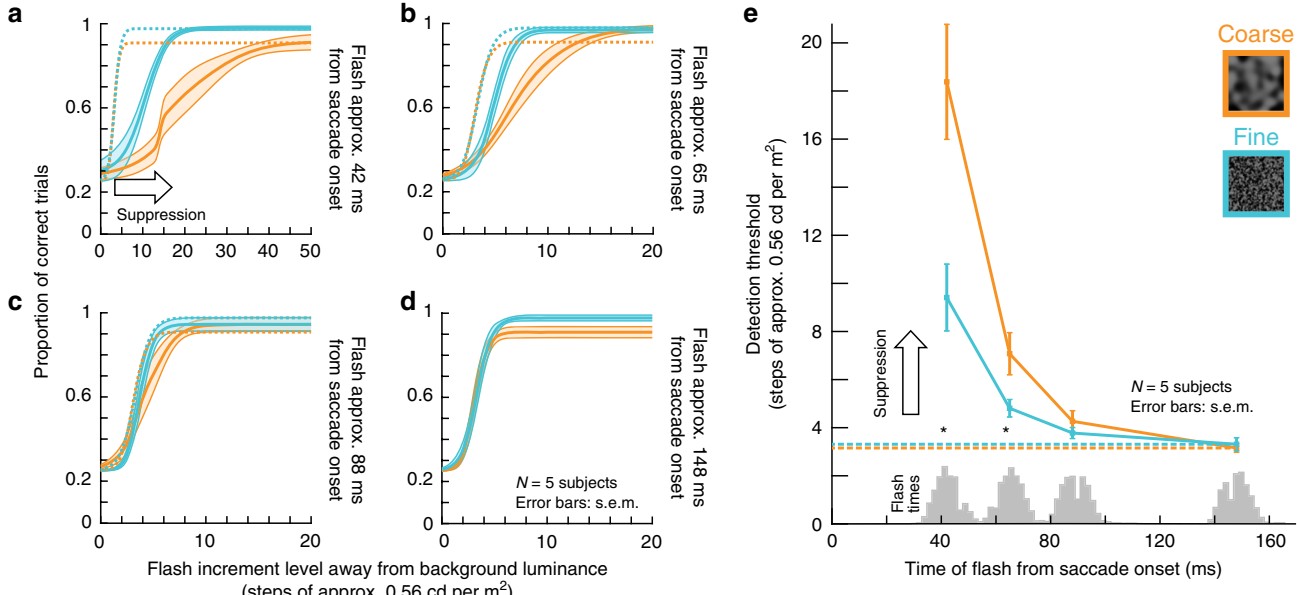

**Fig. 2 Image-dependent elevation of perceptual thresholds across saccades. a–d** Full psychometric curves of flash visibility with the paradigm like in Fig. 1. Solid curves: mean ± s.e.m of the individual psychometric curves of $N = 5$ subjects (individual results in Supplementary Fig. 4). Dashed curves: psychometric curves near recovery from suppression long after saccades (same data as in **d**). Orange and light-blue: data for coarse and fine textures, respectively. **a** For flashes approximately 42 ms from saccade onset, strong perceptual saccadic suppression occurred (compare solid with dashed curves), and coarse textures yielded stronger perceptual saccadic suppression than fine textures. **b** At approximately 65 ms after saccade onset, substantial recovery was visible (note the different x-axis scale from **a**), still with stronger suppression for coarse than fine textures. **c, d** Recovery of visibility continued at later times after saccade onset (88 ms, **c**, and 168 ms, **d**), consistent with Fig. 1. **e** Perceptual detection thresholds (i.e., flash luminance levels needed to achieve a certain correct performance rate; Methods) from **a–d** as a function of flash times from saccade onset. Since flash times were determined using online saccade detection, there was some variability of actual displayed flash times; the gray histograms on the x-axis show the actual distributions of flash times for each group of data from **a–d**. Asterisks denote significant ($p < 0.05$) differences between coarse and fine textures (two-tailed two-sample t-test). Exact p-values at each flash time: 42 ms ($p = 0.012$) and 65 ms ($p = 0.044$). The dashed horizontal lines show the detection thresholds at the longest flash times (**d**); note that these thresholds are also similar to those in the control experiments for visibility (Supplementary Fig. 3a, b).

and it looks qualitatively indistinguishable from saccadic suppression at downstream neural sites[13,14,41] and, indeed, perception (Figs. 1 and 2).

Importantly, retinal "saccadic suppression" strongly depended on background texture (Fig. 3e), exactly like in perception (Figs. 1 and 2). Specifically, we quantified retinal "saccadic suppression" by calculating a neuronal modulation index, defined as $(r_d - r_b)/(r_d + r_b)$. $r_d$ is the response strength to the probe flash presented with a delay $d$ relative to the texture displacement onset, and $r_b$ is the baseline response strength (Methods). The great majority of RGCs were strongly suppressed during and after texture displacements (indicated by negative modulation indices), with gradual recovery afterwards (Fig. 3e; Supplementary Fig. 5 shows the underlying population data), and suppression was more pronounced for coarse than fine textures (Fig. 3e and Supplementary Fig. 5). These results are consistent with the dependence of human perceptual saccadic suppression on background texture statistics (Figs. 1 and 2), suggesting that this dependence starts already in the retina.

We also found that retinal "saccadic suppression" was a robust phenomenon across many different RGCs with diverse properties (Supplementary Fig. 6). Further, it occurred both in mouse (Fig. 3e, left) and pig (Fig. 3e, right) retinae, two mammalian species with different native oculomotor behavior, different lifestyles, and different eye sizes. Thus, our results so far suggest that perceptual saccadic suppression (Figs. 1 and 2), including its dependence on background texture statistics, most likely originates in the retina (Fig. 3), being the outcome of very general retinal-circuit mechanisms that are conserved across species.

**Stimulus–stimulus interactions underlie retinal suppression.** To understand the underlying mechanisms for "saccadic suppression" in the retina in more detail, we explored its properties using different analyses and additional stimulus manipulations. First, we wondered about neural activity saturation, given that saccade-like texture displacements before flash onset could activate RGCs (e.g., Fig. 3b). Specifically, if RGC activity is elevated by the texture displacement, then any subsequent flash-induced response could have caused the cell to reach activity saturation. However, this was not sufficient to explain our results. For example, we observed that suppression often also occurred in RGCs that did not respond strongly to the texture displacements in the first place (Fig. 4a).

Second, we investigated whether retinal "saccadic suppression" critically depended on particular saccade-like speed profiles. In the original experiments (Fig. 3), we simulated saccade-induced image translation speeds to the best of our abilities (given display refresh rates; Methods). However, if we replaced the original translation over 100 ms with a sudden texture jump in one display update (an infinite-speed texture jump), then the same suppression took place, with similar dependence on texture statistics (Fig. 4b). Similarly, in yet another manipulation, we presented the probe flash before the texture displacement; the second response (now to the texture displacement) was suppressed (Fig. 4c). This suggests that retinal "saccadic suppression" can be explained by general stimulus–stimulus interaction effects. As a result, it is a phenomenon that is unlikely to critically depend (qualitatively) on the specific oculomotor repertoire of either mice, pigs, or humans.

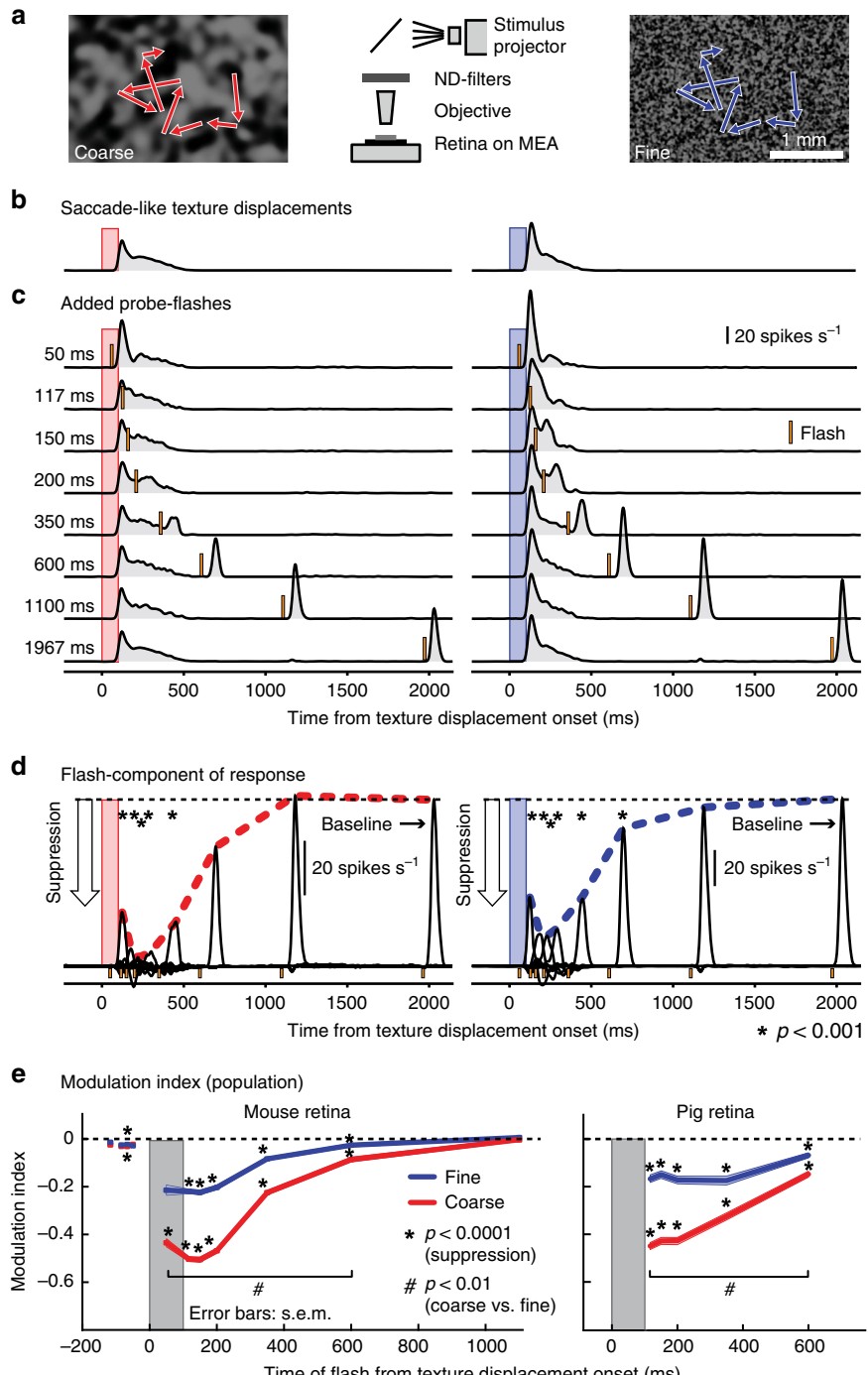

**Fig. 3 "Saccadic suppression" in retina. a** We recorded RGC activity from ex vivo retinae placed on multi-electrode arrays (MEA). A coarse (left) or fine (right) texture was repeatedly translated in a saccade-like manner (red or blue scan paths), and we presented brief visual flashes at different times relative to "saccades" (similar to Fig. 1). **b, c** Average activity of an example RGC to 39 texture displacements alone (**b**) or followed by probe flashes at different time delays (**c**). Red and blue bars show the timings of the texture displacements; orange bars indicate probe flashes. Flash-induced responses were strongly suppressed immediately following saccade-like texture displacements. **d** Isolated flash responses of the same RGC obtained by subtracting responses in **b** from those in **c**. Dashed colored lines highlight the time courses of retinal "saccadic suppression" relative to baseline flash-induced responses. Asterisks indicate flash-induced responses that are significantly suppressed from baseline ($p < 0.001$, one-tailed sign test; Methods). **e** Population modulation index (mean ± s.e.m.) across individual RGCs highlighting retinal "saccadic suppression" (Methods; negative values indicate suppressed flash-induced neural responses). Both mouse and pig retinae showed strong suppression during and after texture displacements, which also depended on texture statistics (similar to perception; Figs. 1 and 2). Asterisks indicate statistically significant suppression ($p < 0.0001$, two-tailed Wilcoxon signed-rank test; Methods). Hash symbols indicate significant differences in suppression between coarse and fine textures ($p < 0.01$, two-tailed Wilcoxon signed-rank test; Methods). Exact $p$-values are indicated in Supplementary Fig. 5. The numbers of recorded cells at each flash time in **e** were: mouse RGCs: $N = 179$ (−177 ms, −84 ms, −50 ms), 161 (−67 ms), 136 (50 ms), 527 (117 ms), 520 (150 ms), 502 (200 ms, 600 ms), 688 (350 ms), 345 (1100 ms); pig RGCs: $N = 228$ for each time point. Supplementary Figs. 5 and 6 show the population data underlying panel **e**.

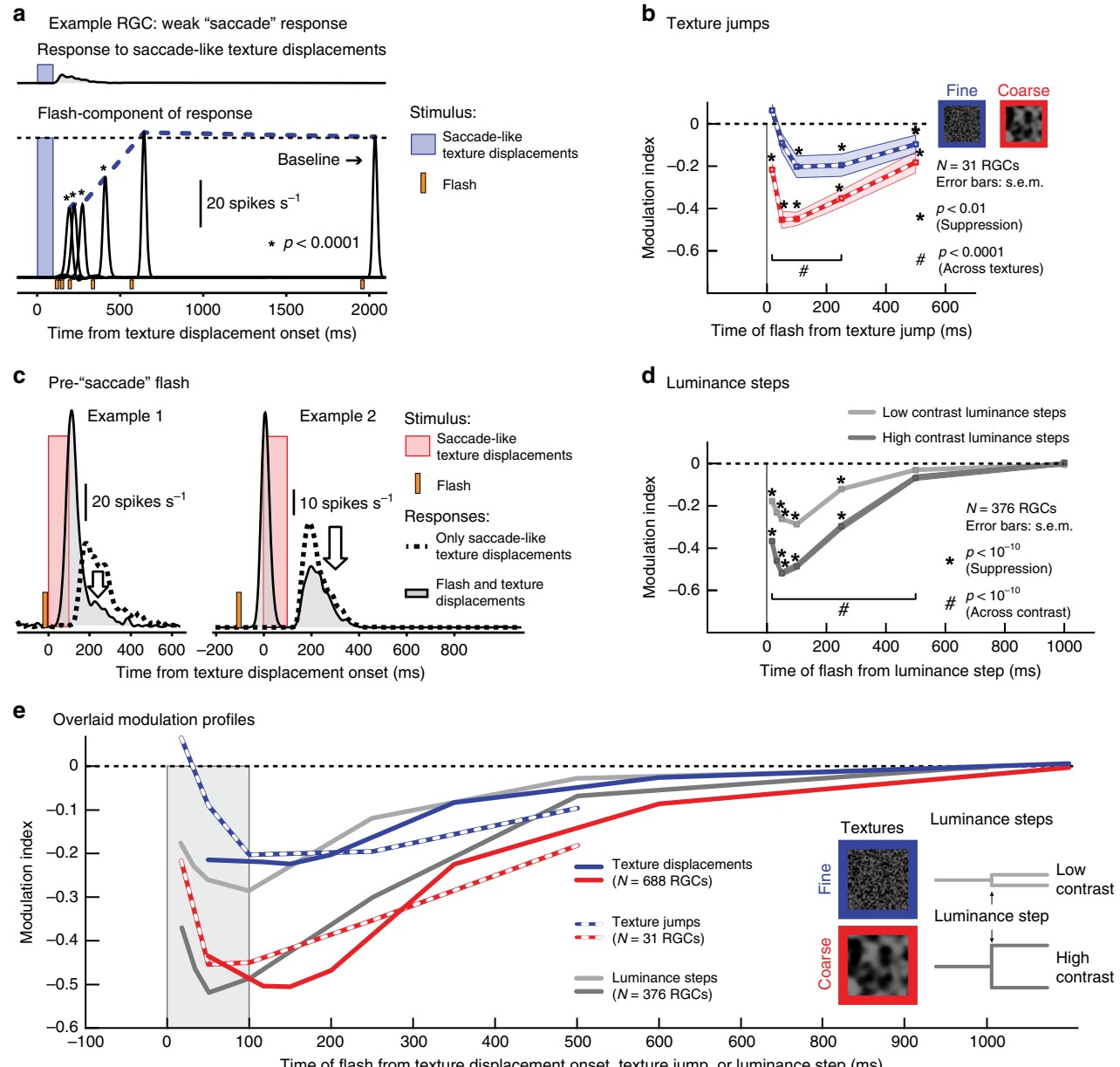

**Fig. 4 Stimulus-stimulus interactions in retinal "saccadic suppression". a** Example RGC responding only weakly to texture displacements (top), but nevertheless exhibiting strong suppression of flash-induced responses (bottom; curves plotted at the same scale). Asterisks indicate significantly suppressed flash-induced responses ($p < 0.0001$, one-tailed sign test, $N = 39$ independent observations; Methods). Exact $p$-values at each flash time: 117 ms ($p = 10^{-5}$), 150 ($10^{-5}$), 200 ($10^{-6}$), 350 ($10^{-5}$), and 600 (0.26). **b** Population modulation index (mean ± s.e.m., $N = 31$ RGCs) when the textures jumped from their start to end positions instantaneously. Strong suppression (*$p < 0.01$, two-tailed Wilcoxon signed-rank test) and significant differences between coarse (red) and fine (blue) textures (#$p < 0.0001$, two-tailed Wilcoxon signed-rank test) still occurred. Exact $p$-values at each flash time (coarse, fine, across): 17 ms ($p = 10^{-6}$, $p = 0.027$, $p = 10^{-6}$), 50 ($10^{-6}$, 0.021, $10^{-6}$), 100 ($10^{-5}$, 0.001, $10^{-6}$), 250 ($10^{-6}$, 0.001, $10^{-4}$), and 500 (0.002, 0.04, 0.06). **c** Two example RGCs showing that a flash before saccade-like texture displacements suppressed the response to the displacements, suggesting that stimulus–stimulus interactions drive retinal "saccadic suppression". **d** Population modulation index (mean ± s.e.m., $N = 376$ RGCs) for a paradigm similar to **b**, but with textures replaced by spatially uniform backgrounds of different intensity (i.e., instantaneous luminance steps). Suppression of flash-induced responses was preserved (*$p < 10^{-10}$, two-tailed Wilcoxon signed-rank test), and differences between low-contrast (light gray) and high-contrast (dark gray) luminance steps (#$p < 10^{-10}$, two-tailed Wilcoxon signed-rank test) resembled the differences between fine and coarse texture jumps in **b**. Exact $p$-values at each flash time (high contrast, low contrast, across contrasts): 17 ms ($p = 10^{-48}$, $p = 10^{-32}$, $p = 10^{-43}$), 33 ms ($10^{-55}$, $10^{-41}$, $10^{-48}$), 50 ms ($10^{-60}$, $10^{-46}$, $10^{-51}$), 100 ms ($10^{-57}$, $10^{-50}$, $10^{-42}$), 250 ms ($10^{-39}$, $10^{-33}$, $10^{-26}$), 500 ms ($10^{-8}$, 0.02, $10^{-8}$) and 1000 ms (0.9, 0.7, 0.8). **e** Overlaid modulation profiles from texture displacements (Fig. 3e), texture jumps (**b**), and contrast steps (**d**). Coarse texture displacements, coarse texture jumps, and high-contrast luminance steps had similar effects; and so did fine texture displacements, fine texture jumps, and low-contrast luminance steps.

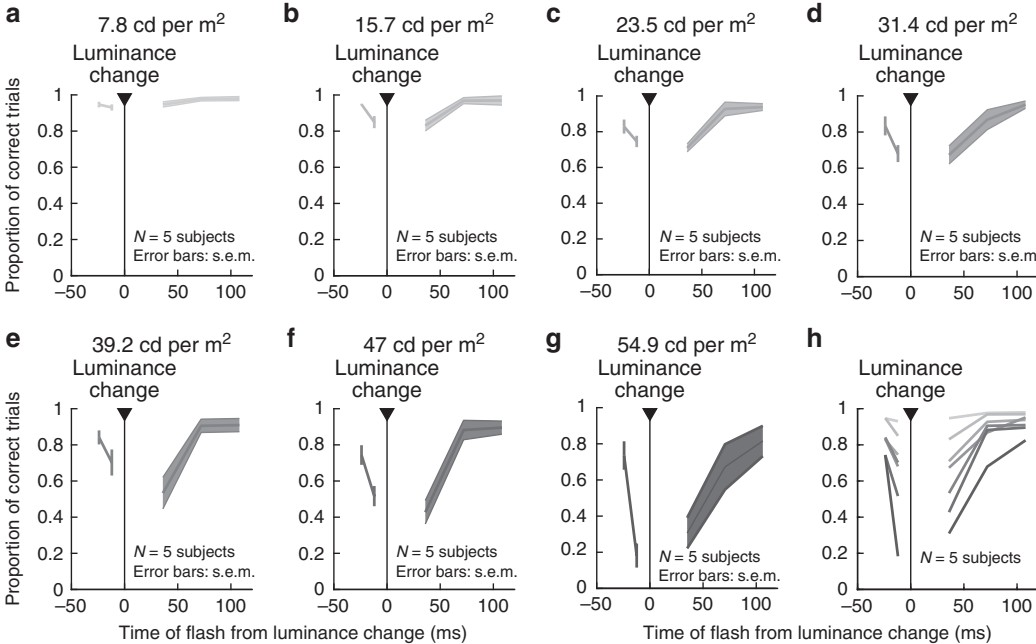

**Fig. 5 Stimulus–stimulus interactions in perceptual suppression without saccades (similar experiment to the retinal paradigm of Fig. 4d).** Subjects fixated and detected brief probe flashes as in the experiments of Figs. 1 and 2; here, the flashes happened around the time of a luminance step (i.e., a sudden change in background luminance) instead of a saccade. The title above each panel indicates the absolute value of the luminance change that took place. **a–g** Proportion of correct responses as a function of flash time from the time of background luminance step. There was progressively stronger perceptual suppression with increasing contrast of the luminance step, consistent with the retinal results of Fig. 4d. **h** Summary of panels **a–g**. Darker colors denote larger absolute values of background luminance changes. Since coarse textures (Figs. 1–4) presumably cause larger contrast variations over retinal receptive fields, this suggests that the image dependence of perceptual saccadic suppression (Figs. 1 and 2) is mediated by stimulus–stimulus interaction effects originating in the retina (Fig. 4d).

The most compelling evidence for stimulus–stimulus interactions underlying retinal "saccadic suppression" came from experiments replacing the texture displacements with structure-free luminance steps (Fig. 4d). Specifically, instead of a background texture that we displaced, we exposed the retina to a uniform gray background and introduced a sudden luminance increase or decrease as the visual transient. This luminance step was either of high (±0.20 to ±0.40 Michelson contrast) or low contrast (±0.03 to ±0.15 Michelson contrast, Methods). The probe flash then followed the luminance step as in the original experiments. Flash responses were indeed suppressed after luminance steps, and this suppression was stronger after high-than after low-contrast visual transients. Interestingly, suppression after high- and low-contrast luminance steps resembled suppression after coarse and fine texture displacements, respectively (e.g., Fig. 3), both in terms of time course and strength (Fig. 4e). Presumably, moving the larger blobs of a coarse texture across the retina would result in high-contrast changes within individual relevant retinal receptive fields (e.g., from a bright blob in a receptive field before texture displacement to a dark blob thereafter), while the smaller blobs in the fine texture would be spatially averaged within receptive fields, resulting in low-contrast changes.

When we next performed human psychophysical experiments mimicking the luminance step retinal experiments, we found remarkably congruent results (Fig. 5). Specifically, subjects maintained saccade-free fixation, and we changed the luminance of the homogenous background (Methods). At random times relative to the change, we presented brief probe flashes like in Fig. 1. All subjects experienced clear perceptual suppression around the luminance steps. Importantly, perceptual suppression depended on the contrast of the luminance change: with a small change in background luminance, suppression was minimal; with

a large change, suppression was strong and long lasting (Fig. 5). As we discuss below, we also observed perceptual suppression even for flashes before the background luminance changes; this matters for interpretations of pre-movement perceptual saccadic suppression (e.g., see Fig. 6 below).

Therefore, the most likely mechanism for retinal "saccadic suppression" is that it emerges as a result of retinal-circuit image processing that is initiated by visual transients; whether they be through texture displacements, infinite-speed texture jumps, or luminance steps (Fig. 4e). It is intriguing that such stimulus–stimulus retinal effects may be inherited deep into the brain's visual processing hierarchy, including cortical (frontal eye field) and subcortical (superior colliculus) areas[42] that are implicated in saccadic suppression[14,41,43,44].

**Motor-related signals shorten visually derived suppression.** In retina, we not only observed similarities to perceptual saccadic suppression (the presence of retinal suppression, and its dependence on texture statistics or luminance step contrast). We additionally noticed that retinal "saccadic suppression" was particularly long lasting (e.g., Fig. 3e). To explore the potential perceptual implications of this observation, we next asked our subjects to maintain fixation while we introduced saccade-like texture displacements in a manner similar to the retinal experiments of Fig. 3 (Fig. 6a); brief flashes occurred around the time of these "simulated saccades" like in Fig. 1. This time, due to the absence of real saccades (trials with microsaccades were excluded), non-visual (motor-related) components could not influence flash-induced neural responses and perception. Still, given the retinal results (Figs. 3 and 4), we had three hypotheses that we validated: (1) strong perceptual suppression still occurred regardless of texture details (Fig. 6b, c);

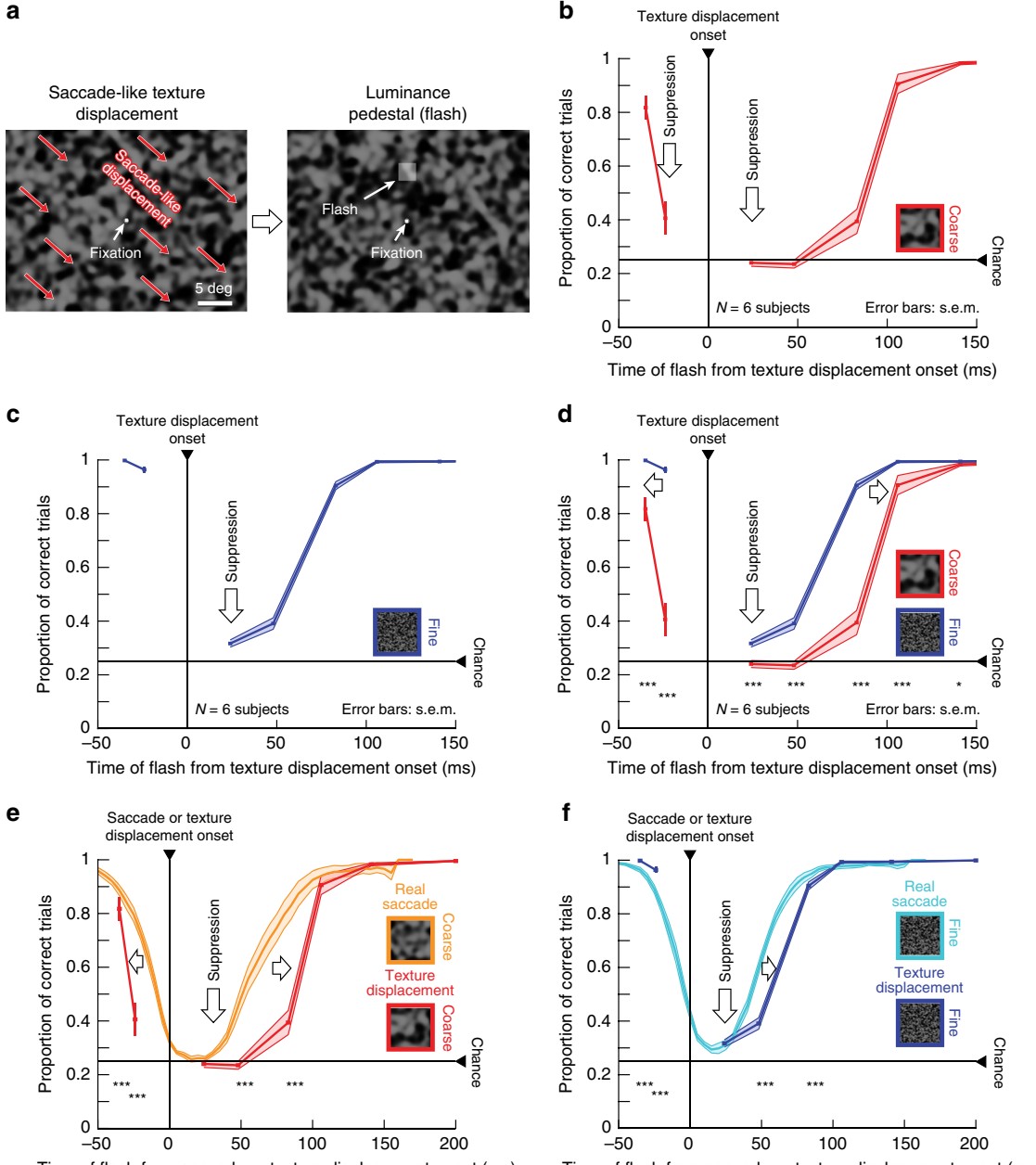

**Fig. 6 Image dependence of perceptual suppression without saccades. a** Rapid texture displacements simulating saccade-like image displacements, similar to the retina experiments (Fig. 3). We used the same flashes and simulated saccade directions as in Fig. 1. The example shows a coarse texture (fine textures shown in insets in **c**, **d**, and **f**). **b**, **c** Pre-, peri-, and post-displacement perceptual suppression (mean ± s.e.m. over $N = 6$ individual subjects) occurred for both coarse (**b**) and fine (**c**) textures. **d** As with real saccades (Fig. 1), suppression started earlier and lasted longer with coarse textures (also compare to similar retinal effects in Fig. 3e). Notably, pre-displacement suppression depended on texture statistics, just like with real saccades (Fig. 1). **e**, **f** Simulated saccades were associated with significantly longer suppression than real saccades for both fine and coarse textures. For coarse textures (**e**, which were most effective in causing suppression overall), flashes presented before the "saccade" event were suppressed earlier in the simulated saccade condition than in the real saccade condition (also see Fig. 7); thus, prolonged suppression with texture displacements was not restricted to post-displacement flashes only. Asterisks denote significant differences between coarse and fine textures (**d**) or between real and simulated saccades (**e**, **f**) at each indicated time point ($\chi^2$ tests with Bonferroni corrections; *$p < 0.005$ in **d** and $p < 0.007$ in **e**, **f**; ***$p < 0.0001$ in **d** and $p < 0.00014$ in **e**, **f**). Supplementary Fig. 2 shows individual subject results.

(2) suppression strength and duration depended on texture statistics (Fig. 6d); and (3) suppression outlasted suppression with real saccades (Fig. 6e, f). This last point, in particular, suggests that motor-related saccadic signals may act to shorten the perceptual interruption resulting from visually induced saccadic suppression, while maintaining the putatively retinally determined (Figs. 3 and 4) dependence on image statistics. Note

also that the first and third points above are consistent with earlier observations by Diamond et al.[16].

In humans, we observed perceptual suppression also prior to saccade-like texture displacements[19,26] (Fig. 6). This was again consistently dependent on texture statistics (Fig. 6b–d; also see Fig. 7 below for additional evidence). Further, like the suppression after saccade onset, this pre-saccadic perceptual suppression

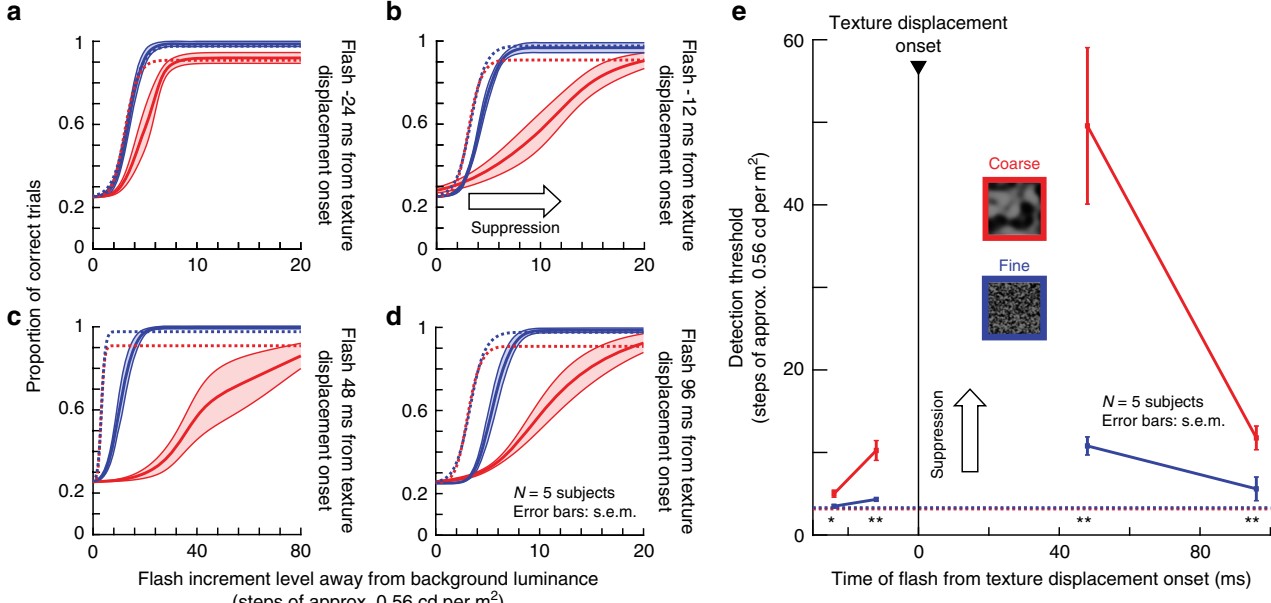

**Fig. 7 Image-dependent elevation of perceptual thresholds without saccades.** Full psychometric curves of flash visibility around the time of simulated saccades (similar to Fig. 2, paradigm similar to Fig. 6). **a–d** Solid curves: mean ± s.e.m of individual psychometric curves of $N = 5$ subjects (individual subject results: Supplementary Fig. 7). Dashed curves: baseline data from the same subjects without simulated saccades and long after any real saccades (same data as in Fig. 2d; also similar to Supplementary Fig. 3a, b with additional subjects). Red and blue: data for coarse and fine textures, respectively. **a** Flash 24 ms before texture displacement onset: coarse texture (red) requires higher flash contrasts (that is, reduced sensitivity) relative to baseline. This effect was much weaker with fine textures (blue). **b** Flash 12 ms before displacement onset: both coarse and fine textures were associated with significant perceptual suppression relative to baseline, consistent with Fig. 6. Suppression was stronger for coarse than fine textures. **c** Perceptual suppression was the strongest (note the different x-axis scale from the other panels) immediately after texture displacement onset. **d** 96 ms after texture displacement onset, there was still significant perceptual suppression, again significantly stronger for coarse than fine textures. This result is consistent with Fig. 6 and highlights the longer-lasting suppression around simulated saccades compared to real saccades (Figs. 1 and 2). **e** Detection thresholds from **a–d** as a function of flash time from texture displacement onset. Pre- and post-displacement perceptual suppression occurred and was stronger with coarse textures. Asterisks: significant differences between coarse and fine textures (two-tailed two-sample t-test; *$p < 0.05$; **$p < 0.01$). Horizontal dashed lines: baseline detection thresholds from Fig. 2d, e. All other conventions are similar to Figs. 1, 2, 6.

was shorter during real saccades than during simulated saccades (due to later onset of suppression, Fig. 6e). Even in our retinal data, we found very slight "pre-saccadic" suppression. However, for retinal responses, the effect size before texture displacements was much smaller than after texture displacements: the strongest "pre-saccadic" retinal effect occurred at −67 ms with a median population modulation index of −0.024 ($p = 6 \times 10^{-8}$, Wilcoxon signed-rank test) compared to −0.55 ($p = 3 \times 10^{-82}$) for "post-saccadic" suppression at 150 ms delay (Fig. 3e and Supplementary Fig. 5b). It is therefore likely that this particular phenomenon, perceptual pre-saccadic suppression (Fig. 6b–f), arises from visual (not movement-command-related) processing further downstream of the retina, perhaps through backwards masking[28,45]. This also holds true for our experiments with background luminance steps (Fig. 5), and it can also explain why peak suppression time in our retinal experiments (Figs. 3 and 4) appeared slightly different from peak suppression time with real saccades (Figs. 1 and 2).

Next, we determined explicit perceptual thresholds for the texture displacement paradigm introduced in Fig. 6, using the QUEST and full psychometric curve procedures described for Fig. 2. We again picked four specific time points relative to texture displacement onset, chosen strategically to highlight perceptual threshold elevations at maximal suppression, to characterize differences in recovery time between coarse and fine textures, and to fill the gap before texture displacement onset. The net conclusion (Fig. 7) was the same as that in Fig. 6. There was robust elevation of perceptual thresholds before, during, and after texture displacements. Most importantly, the elevation was much

stronger and longer-lasting (both before and after texture displacements) for coarse than for fine textures. The effect was also robust across individual subjects (Supplementary Fig. 7).

Therefore, our long-lasting RGC suppression effects (Figs. 3 and 4) were not an idiosyncrasy of our ex vivo electrophysiological procedures, but they were reflected in the longer duration of perceptual suppression after simulated saccades. Importantly, they were indicative of a potential shortening of visually derived suppression in association with real saccades.

**Visually derived suppression underlies even more phenomena.** Our results so far suggest that visual contributions can go a long way in explaining perceptual properties of saccadic suppression (e.g., the presence of suppression, and the dependencies on image content), without the need for invoking mechanisms related to motor commands. We wondered whether visual contributions can also explain classic suppression phenomena in experiments when uniform, rather than textured, backgrounds are used. One such robust phenomenon has been the selective suppression of low spatial frequencies. In a classic study[10], subjects viewed briefly flashed Gabor gratings over a uniform background. Around the time of saccades, visibility of low spatial frequency gratings was suppressed more strongly than of high-frequency gratings. This was interpreted as a motor-related influence on magnocellular pathways[16,17]. Still, convincing neural mechanisms for this phenomenon remain elusive[14,21,29,30,46–50]. Can the strong prominence of visual contributions to saccadic suppression revealed by our results also be extended to account for this

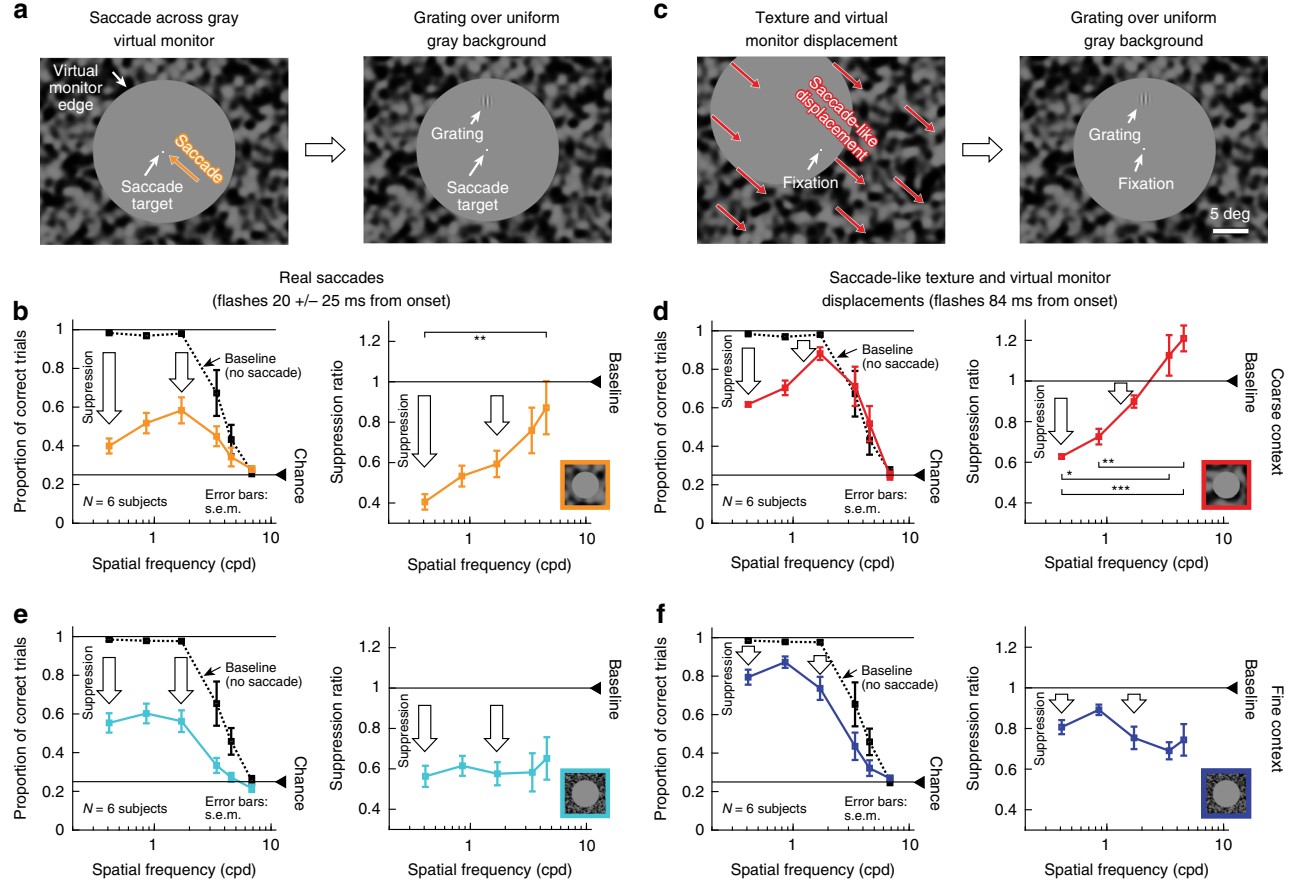

**Fig. 8 Selective peri-saccadic suppression of low spatial frequencies[10] is a visual phenomenon. a** Left: subjects made saccades towards display center. Right: gratings were flashed peri-saccadically over a uniform gray background (circular "virtual monitor" surrounded by a coarse texture; saccade directions and flash locations: similar to Figs. and 6). **b** Left: proportion of correct grating localizations with different spatial frequencies during fixation ("Baseline"; dashed curve) and for peri-saccadically flashed gratings (solid curve). Low spatial frequencies were associated with the strongest suppression relative to baseline. Right: ratio of peri-saccadic to baseline performance (highest spatial frequency not shown because it was at chance performance even in baseline). Suppression depended on grating spatial frequency ($\chi^2 = 13.46$, $p = 0.0092$, $df = 4$, Kruskal–Wallis test; **$p < 0.01$ for post-hoc pairwise comparisons between the lowest and highest spatial frequencies). **c** Left: simulated saccade-induced image displacements by translating the virtual monitor and surrounding texture from one corner towards display center. Right: gratings appeared as in **a**. **d** The same selective suppression of low spatial frequencies occurred as with real saccades (**b**). "Baseline" in this context means both no saccades and no virtual monitor and texture displacements. Suppression depended on spatial frequency ($\chi^2 = 25.33$, $p < 0.0001$, $df = 4$, Kruskal–Wallis test; *$p < 0.05$, **$p < 0.01$, ***$p < 0.001$ for post-hoc pairwise comparisons between individual spatial frequencies). **e**, **f** With a fine surround texture, both real (**e**) and simulated (**f**) saccades were associated with suppression for all spatial frequencies; suppression selectivity[10] was eliminated ($\chi^2 = 0.8$, $p = 0.938$, $df = 4$ for **e** and $\chi^2 = 7.74$, $p = 0.102$, $df = 4$ for **f**, Kruskal–Wallis test). Error bars: s.e.m. across individual subjects' curves. Supplementary Figs. 8–10: full time courses and controls with black surrounds around the virtual monitor. Note that in **d**, **f**, we exploited the longer time course of visual suppression (Fig. 6 and Supplementary Figs. 8 and 9) to probe perception at a later time than in **b**, **e**. This also explains why suppression appeared quantitatively weaker in **d**, **f** than in **b**, **e**.

classic phenomenon? In other words, is peri-saccadic selective suppression of low spatial frequencies[10] fundamentally a visual, rather than motor, phenomenon?

We considered this phenomenon from the perspective of visual input during such experiments: saccades across a uniform background invariably involve moving the image of the video monitor (or other form of display) in visual coordinates. Therefore, the image of any edge discontinuity associated with the display monitor (or with the surrounding cardboard paper around it[10]) will invariably move across the retina. This allows us to ask if one can replicate selective suppression of low spatial frequencies[10] without any saccades at all, solely based on the visual flow during such experiments.

We first replicated the classic phenomenon itself (Methods). Subjects localized briefly flashed vertical Gabor gratings with different spatial frequencies; the flashes occurred peri-saccadically as in Fig. 1a. Here, however, the screen was homogeneous, like in

the classic experiment, with the exception of a surround region showing a stationary texture (the coarse texture used in our earlier experiments, Fig. 8a). We call the large homogeneous region (diameter: 20 deg) the "virtual monitor". The outcome confirmed the classic findings: Fig. 8b (left) shows localization performance for flashed gratings around saccade onset, compared to flashes without saccades (and without any other display transients), and Fig. 8b (right) plots the ratio of those percepts. Perception of low spatial frequency gratings was selectively suppressed (relevant statistics are shown in Fig. 8; full time courses of these effects are shown in Supplementary Figs. 8 and 9). These results are consistent with the classic phenomenon[10].

The presence of the textured surround allowed us to next isolate the effects of visual flow. In separate trials, subjects fixated, and we presented saccade-like image motion. For example, in order to simulate a real saccade from the lower right corner to display center (Fig. 8a), the virtual monitor moved together with

its textured surround from the top left corner towards display center (Fig. 8c). We then briefly presented the same Gabor gratings as in Fig. 8a, b. Relative to fixation position, this experiment was comparable to the situation with real saccades: there was a uniform background against which a brief Gabor grating was flashed. And, indeed, we observed the same selective suppression of low spatial frequencies despite the absence of saccades (Fig. 8d). Moreover, again consistent with our results from Figs. 1–7, the suppression lasted longer than with real saccades (robust selective suppression in Fig. 8d occurred even 84 ms after simulated saccades; Supplementary Figs. 8 and 9). Similar results were obtained with a uniform black surround around the virtual monitor, as might be the case in typical laboratory settings (Supplementary Fig. 10). Therefore, visual mechanisms account even for the results of Burr et al.[10] and similar experiments[14] using uniform backgrounds, without the need to invoke non-visual (motor-related) mechanisms.

Motivated by the differences between coarse and fine textures in Figs. 1–7, we next replaced the coarse texture around the virtual monitor (Fig. 8c) with a fine texture, and we repeated the experiments with simulated saccades (Fig. 8f). Surprisingly, we observed uniform suppression for all spatial frequencies (Fig. 8f). In other words, the specific suppression of low spatial frequencies (Fig. 8c, with saccade-like visual flow, but without eye movements) depended on the visual context containing a coarse texture in the visual surround. This led to a very strong prediction: if saccadic suppression properties do indeed rely on visual processing, then suppression during real saccades should depend mainly on visual context; one should be able to easily violate the classic phenomenon (namely, the specific suppression of low spatial frequencies[10]). This is exactly what we found (Fig. 8e): for real saccades across the virtual monitor, and with the surrounding visual context being a fine rather than coarse texture, we observed perceptual suppression for all gratings, abolishing suppression selectivity for low spatial frequencies. In all cases, the effects were not explained by motor variability across surround texture conditions (Supplementary Fig. 3e, f).

We further confirmed all these observations by collecting full psychometric curves (Methods), similar to Figs. 2 and 7 above: Fig. 9 shows results for real saccades, and Fig. 10 for simulated saccades. In both cases, with a coarse surround texture, perceptual threshold was elevated more strongly for low spatial frequency Gabor patches. With a fine surround texture, perceptual threshold was elevated non-specifically for all probe Gabor patches.

In summary, perceptual saccadic suppression occurred in all of our experiments, either with or without real saccades, simply as a function of visual flow (Figs. 1, 2, 6–10). Simple visual transients, without the need for saccade-like stimulus kinematics, were sufficient to elicit suppression in both retina and perception (Figs. 4 and 5). Such suppression quantitatively depended on scene statistics, both for full-field textures (Figs. 1, 2, 6, 7) in a manner predicted by retinal processing (Figs. 3–5), and for textures limited to the surround (Figs. 8–10). Even the suppression selectivity of low spatial frequency Gabor probes[10] was determined by visual context (Figs. 8–10).

## Discussion

We found that visual image processing accounts for a large component of classic perceptual demonstrations of saccadic suppression, and that such image processing occurs as early as in the very first stage of visual processing, the retina. In fact, we found remarkable congruence between the image dependence of three seemingly disparate phenomena: perceptual suppression with real saccades (Figs. 1 and 2), perceptual suppression with simulated saccades (Figs. 6 and 7), and neural suppression in

RGCs, which carry the retinal output (Figs. 3 and 4). In all cases, modifying the background texture statistics resulted in highly predictable changes in suppression profiles. This was further corroborated in both the retina (Fig. 4d) and perception (Fig. 5) when we replaced texture displacements with simple background luminance steps.

Key to all our observations is the single insight that, from the perspective of visual image processing, a saccade itself generates a potent visual stimulus. For example, our RGCs often responded vigorously to saccade-like image displacements (Fig. 3b). Therefore, when probing peri-saccadic perceptual sensitivity using brief flashes, as in classic studies of perceptual saccadic suppression, the visual system is not only responding to the externally provided flashes, but it is also responding to the self-induced visual flows caused by eyeball rotations. These saccade-induced retinal image shifts trigger visual mechanisms that can suppress the retinal response to subsequent stimulation. Such suppression is not exclusive to saccades. It instead occurs for any scenario that involves sequential visual stimulation, including visual masking[2,27,28,45] and double-flash[42] paradigms. It is, therefore, not surprising that the outcome is also comparable: the response to a second stimulus is suppressed by the presence of a first stimulus, be it a mask, a flash, or transients caused by saccades. Indeed, our own results demonstrate that simpler sequential visual stimulation with luminance steps plus probe flashes shows qualitatively similar perceptual (Fig. 5) and retinal (Fig. 4d) suppression profiles to those seen with simulated saccades. Therefore, classic saccadic suppression paradigms, employing brief peri-saccadic visual probes, are essentially stimulus–stimulus paradigms from the perspective of visual flow on the retina.

Additional support for the above sentiment emerges from the time courses of stimulus–stimulus neural adaptation effects in areas like the frontal eye field and superior colliculus[42]. These time courses are particularly intriguing because they agree with our observations that retinal (Figs. 3 and 4) and perceptual (Figs. 6 and 7) suppression with simulated saccades had longer suppression time courses than observed with real saccades (Figs. 1 and 2). Indeed, the time courses of the neural adaptation effects in the frontal eye field and superior colliculus[42], and related brain areas, are similar to our observed perceptual time courses without real saccades. Given that both the frontal eye field and superior colliculus have previously been implicated in saccadic suppression[14,41,43,44], it is thus conceivable that suppression in these areas is inherited, at least partially, from the retina.

Looking forward, it is imperative to investigate the neural mechanisms behind visual masking in much more detail. In our perceptual experiments with simulated saccades (Figs. 6 and 7), we saw clear suppression even with probe flashes before texture displacement. That is, perceptual localization of the probes was masked, backwards in time, by the subsequent texture displacement. In the past, pre-saccadic suppression with real saccades (e.g., Fig. 1) was sometimes taken as evidence that perceptual saccadic suppression is fundamentally driven by motor-related signals like corollary discharge. However, our results (Figs. 6 and 7) show that a visual transient is sufficient. Even simple background luminance steps were associated with pre-step perceptual suppression (Fig. 5). These effects have been described as backwards visual masking[45], but what are the underlying neural mechanisms? Such backwards masking was not present in our retinal results, certainly not as clearly as in perception, so it must emerge through visual mechanisms in other brain structures.

One possibility could be related to the fact that priors strongly influence the perceptual interpretation of sensory evidence. In the case of global retinal image motion, which is caused by eye movements in most real-world scenarios, priors could influence the percept of a flash occurring before a saccade or texture

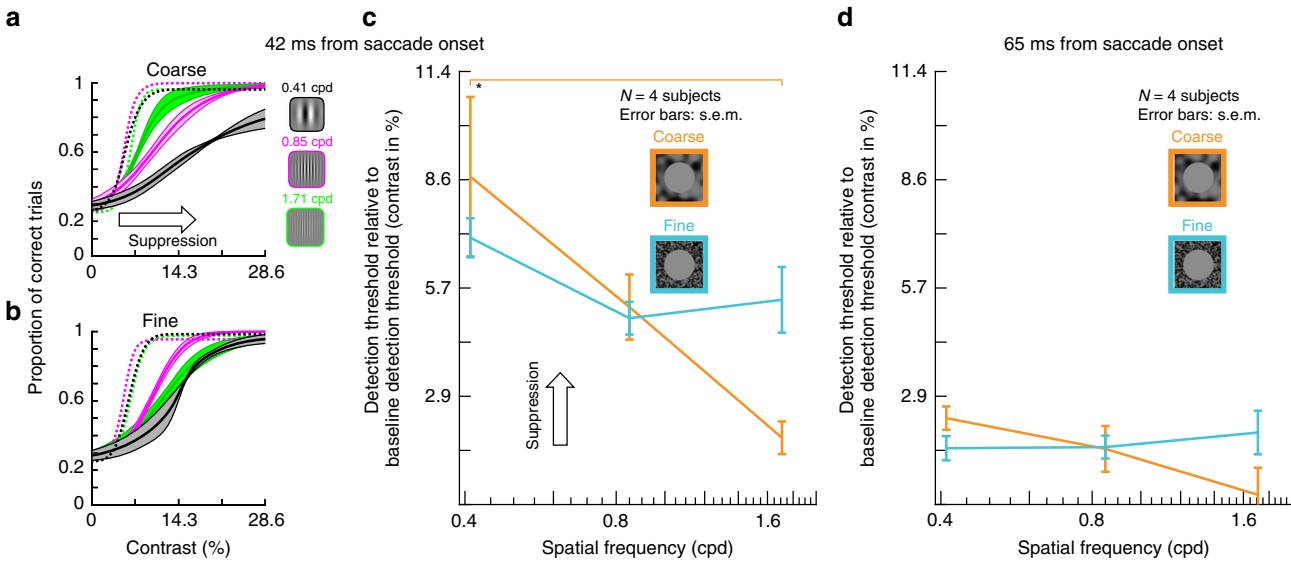

**Fig. 9 Selective and unselective saccadic suppression measured using full psychometric curves. a** We repeated the real saccade experiments of Fig. 8, and obtained full psychometric curves by using different Gabor grating contrasts (Methods). Different colors indicate different spatial frequencies of the flashed gratings. When the gratings were flashed ~42 ms after saccade onset (Methods) and there was a coarse surround texture, perceptual suppression clearly depended on spatial frequency: detection thresholds were highest for the lowest spatial frequency, and they progressively decreased with increasing spatial frequency. Each curve shows mean ± s.e.m. of four subjects' psychometric curves. Dashed psychometric curves show perceptual detectability without saccadic suppression (obtained similarly to Fig. 8). **b** When the surround context was fine, rather than coarse, perceptual suppression was not selective for low spatial frequencies (consistent with Fig. 8). **c** Detection thresholds from **a**, **b** as a function of grating spatial frequency for flashes ~42 ms after saccade onset. With a coarse surround, detection thresholds were highest for low spatial frequencies and progressively decreased with increasing spatial frequency (1-way ANOVA, $p = 0.0168$, $F = 6.6608$; $p = 0.0133$ for post-hoc comparison between lowest and highest spatial frequency, indicated by *). With a fine surround, detection thresholds did not depend on spatial frequency. **d** Same as in **c** but now for grating flashes occurring ~65 ms after saccade onset. For both surround textures, detection thresholds decreased, indicating perceptual recovery. There was still a trend for dependence of perception on spatial frequency in the coarse condition, consistent with **c**.

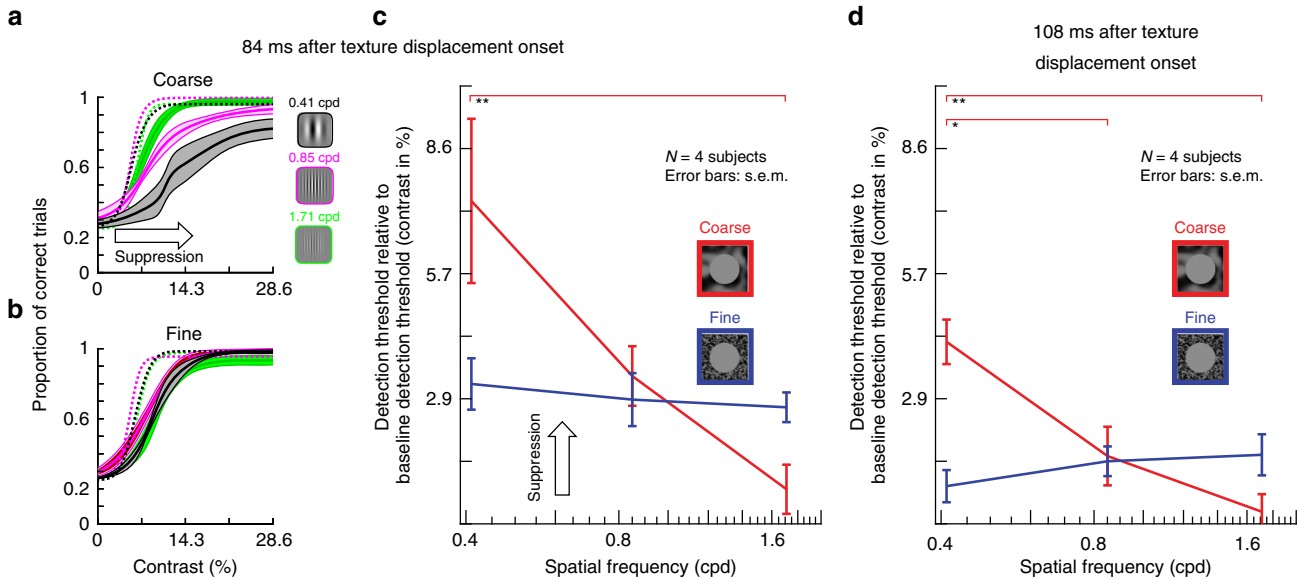

**Fig. 10 Selective and unselective saccadic suppression without any saccades.** This figure is identical to Fig. 9, except that real saccades were replaced (in the same subjects) with simulated saccades (exactly as in Fig. 8). All of the same conclusions were reached. There was selective suppression for low spatial frequencies when the texture surround was coarse (**a**); suppression was unselective for grating spatial frequency with a fine surround (**b**); and there was gradual recovery with time (**c**, **d**). In fact, perceptual suppression was clearer and longer lasting in this condition than with real saccades (also consistent with Figs. 1, 6, 8). All other conventions are as in Fig. 9. In **c**, the coarse texture surround showed a significant main effect of spatial frequency (1-way ANOVA, $p = 0.0113$, $F = 7.6878$; $p = 0.0092$ for post-hoc comparison between lowest and highest spatial frequency, indicated by **). In **d**, the coarse surround also showed a significant main effect of spatial frequency (1-way ANOVA, $p = 0.0019$, $F = 13.5276$; $p = 0.0017$ for post-hoc comparison between lowest and highest spatial frequency, and $p = 0.0186$ for post-hoc comparison between lowest and intermediate spatial frequency).

displacement. Specifically, such priors may cause perception to "omit" the pre-saccadic flash even though it evokes a strong retinal transient. This would happen exactly because of the pairing of the flash with a very likely occurrence of a saccade, interpreted as such due to the global image motion, even if its neural transient in the retina is weakened by the prior flash. This would result in a kind of credit assignment problem due to a strong prior association of global image motion with saccades.

More generally, our results suggest that visual flow is important in perceptual saccadic suppression, even in paradigms that have often been taken as indication for motor-related top-down suppression (Figs. 8–10). It would be interesting in the future to further test the generalizability of this notion. We were indeed greatly surprised when we found that selective suppression of low spatial frequencies[10] can be violated in two important ways. First, the suppression selectivity can be abolished with a simple change of visual context. Second, the same selective suppression of low spatial frequencies can be obtained without saccades. Thus, with or without saccades, either selective or nonselective suppression could occur as a function of visual flow. In hindsight, this might shed light on a surprising recent finding in superior colliculus[14]. There, using essentially the same paradigms, only one type of superior colliculus visually responsive neurons (so-called visual-motor neurons) exhibited selective suppression of low spatial frequency sensitivity. The other type of visually responsive neurons (visual-only neurons) showed mild but, critically, non-selective suppression[14]. These two types of neurons occupy different superior colliculus laminae and have different patterns of lateral interactions from across this structure's visual field representation[51]. It is now conceivable, considering our current results (Figs. 8–10), that both patterns of suppression (selective or not) may be embedded simultaneously in these different neuronal populations with specific circuitry and tuning for visual peripheral contexts.

Finally, motor-related mechanisms still likely play an important role in perceptual saccadic suppression. Such mechanisms appear to shorten suppression originating from visual processing (Fig. 6), and might therefore minimize the duration of saccade-induced disruptions. Indeed, a variety of cortical areas exhibit post-saccadic excitability enhancement[52–54]. It would be interesting to further investigate how such enhancement may contribute to the shortened time courses of perceptual saccadic suppression that we observed (e.g., Fig. 6e, f). Furthermore, besides just suppression, saccades are also associated with "omission", the lack of awareness of intra-saccadic background image motion[22,55]. It would, therefore, also be interesting to study the neural mechanisms through which strong saccade-induced neural transients in the retina (Fig. 3b) are perceptually "omitted" to give the illusion of continuous perception across saccades. More intriguingly, saccades also cause spatial updating of visual reference frames (compensating the image shifts that they cause). Information contained in the motor command itself is likely critical for adjusting spatial receptive fields across saccades, as observed in some brain areas[56,57]. Our findings leave open the possibility, however, that trans-saccadic image flow might play a role in this phenomenon as well.

## Methods

**Ethics approvals**. We performed electrophysiological experiments on ex vivo mouse and pig retinae as well as non-invasive perceptual experiments on human subjects.

Animal use was in accordance with German and European regulations, and animal experiments were approved by the Regierungspräsidium Tübingen.

Human subjects provided written, informed consent, and they were paid 8–15 Euros per session of 45–90 min each. Depending on the experiment, each subject was measured for 2–10 sessions (detailed trial and session numbers are provided below). Human experiments were approved by ethics committees at the Medical

Faculty of Tübingen University, and they were in accordance with the Declaration of Helsinki.

**Retina electrophysiology laboratory setup**. We used retinae extracted from *PV-Cre x Thy-S-Y* mice (*B6;129P2-Pvalbtm1(cre)Arbr/J × C57BL/6-tg (ThystopYFPJS)*), which are functionally wild type[58–60]. Twenty-three retinae from seven male and fifteen female mice (3–12-months-old) were used. We also replicated experiments on pig retina obtained from domestic female pigs after they had been sacrificed during independent studies at the Department of Experimental Surgery in our Medical Faculty. We used nine pig retinae.

We housed mice on a 12/12 h light/dark cycle in ambient temperature, ranging between 20–22 °C, and humidity levels of ~40%. Mice were dark adapted for 4–16 h before experiments. We then sacrificed them under dim red light, removed the eyes, and placed eyecups in Ringer solution (in mM: 110 NaCl, 2.5 KCl, 1 CaCl$_2$, 1.6 MgCl$_2$, 10 D-glucose, and 22 NaHCO$_3$) bubbled with 5% CO$_2$ and 95% O$_2$. We removed the retina from the pigment epithelium and sclera while in Ringer solution.

Pigs were anesthetized using atropine, azaperone, benzodiazepine (midazolam), and ketamine, and then sacrificed with embutramide (T61). Before embutramide administration, heparin was injected. The pigs were dark adapted for 15–20 min before sacrifice. Immediately after sacrifice, the eyes were enucleated under dim red light, and the cornea, lens, and vitreous were removed. Eyecups were kept in CO$_2$-independent culture medium (Gibco) and protected from light. We transported eyecups to our laboratory and cut pieces from mid-peripheral or peripheral retinae. Only those retinae that were healthy and showed ganglion cell responses to light stimuli were used in our experiments.

We recorded retinal ganglion cell (RGC) activity using either low- or high-density multi-electrode arrays (MEAs). The low-density setup consisted of a perforated 60-electrode MEA (60pMEA200/30ir-Ti-gt, Multichannel Systems, Reutlingen, Germany) having a square grid arrangement and 200 μm inter-electrode distance. We mounted an isolated retina on a nitrocellulose filter (Millipore) with a central 2 × 2 mm hole. The mounted retina was placed with the RGC-side down into the recording chamber, and good electrode contact was achieved by negative pressure through the MEA perforation. We superfused the tissue with Ringer solution at 30–34 °C during recordings, and we recorded extracellular activity at 25 kHz using a USB-MEA-system (USB-MEA 1060, Multichannel Systems) or a memory-card based system (MEA1060, Multichannel Systems). Data were acquired using MC Rack version 4.6.2 (Multichannel Systems). More details are provided in Reinhard et al.[61].

The high-density MEA setup consisted of either a HiDens CMOS MEA[62] (developed by the lab of Andreas Hierlemann, Basel, Switzerland) or a MaxOne system[63] (Maxwell Biosystems, Basel, Switzerland). The HiDens CMOS MEA featured 11,011 metal electrodes with inter-electrode (center-to-center) spacing of 18 μm placed in a honeycomb pattern over an area of 2 × 1.75 mm. Any combination of 126 electrodes could be selected for simultaneous recording. The MaxOne MEA featured 26,400 metal electrodes with center-to-center spacing of 17.5 μm over an area of 3.85 × 2.1 mm. In this system, up to 1024 electrodes could be selected for simultaneous recordings. For each experiment, a piece of isolated retina covering almost the entire electrode array was cut and placed RGC-side down in the recording chamber. We achieved good electrode contact by applying pressure on the photoreceptor side of the retina by carefully lowering a transparent permeable membrane (Corning Transwell polyester membrane, 10 μm thick, 0.4 μm pore diameter) with the aid of a micromanipulator. The membrane was drilled with 200 μm holes, with center-center distance of 400 μm, to improve access of the Ringer solution to the retina. We recorded extracellular activity at 20 kHz using FPGA signal processing hardware. In the case of the HiDens CMOS MEA, data were acquired using custom data acquisition software, called MEA 1k Scope (developed by the lab of Andreas Hierlemann, Basel, Switzerland). In the case of the MaxOne MEA, data were acquired using MaxLab software provided by Maxwell Biosystems, Basel, Switzerland.

In total, we performed 36 recordings, 24 from mouse and 12 from pig retina. Fifteen of the 36 recordings were done using low-density MEAs. Once a basic experimental protocol was established, we shifted to HiDens CMOS MEA providing much higher throughput. Twelve experiments were done using this setup. We upgraded to the MaxOne MEA for even higher throughput and did our final nine recordings using this setup.

We presented light stimuli to the retinal piece that was placed on the MEA using a DLP projector running at 60 Hz (Acer K11 for low-density MEA experiments and Lightcrafter 4500 for high-density MEA experiments). In all, 60 Hz is above the flicker fusion frequency of both mouse and pig retinae; therefore, the framerate of these projectors was adequate for our purposes. The Acer K11 projector had a resolution of 800 × 600 pixels covering 3 × 2.25 mm on the retinal surface. Lightcrafter 4500 had a resolution of 1280 × 800 pixels, extending 3.072 × 1.92 mm on the retinal surface. We focused images onto the photoreceptors using a condenser (low-density MEA recordings, illumination from below) or a 5x objective (high-density MEAs, illumination from above). In each case, the light path contained a shutter and two motorized filter wheels with a set of neutral density (ND) filters (Thorlabs NE10B-A to NE50B-A), having optical densities from 1 (ND1) to 5 (ND5). Light intensity was adjusted to be in the mesopic range.

We measured the spectral intensity profile (in $\mu W\,cm^{-2}\,nm^{-1}$) of our light stimuli with a calibrated USB2000 + spectrophotometer (Ocean Optics) and converted the physical intensity into a biological equivalent of photoisomerizations per rod photoreceptor per second ($R^*\,rod^{-1}\,s^{-1}$), as described before[60]. Light intensities of the projector output covered a range of 3 log units (i.e., 1,000-fold difference between black and white pixels, over an 8-bit range). We linearized the projector output, and we used only grayscale images of limited contrast, spanning at most the range from 0 to 120 in the 8-bit range of the projector (see stimulus description below for details). Absolute light intensities were set to the mesopic level, where a stimulus intensity of "30" in our 8-bit DLP projector scale (0-255) corresponded to 225 to 425 $R^*\,rod^{-1}\,s^{-1}$, depending on the experimental rig used for the experiment (i.e., different DLP projectors and MEAs). We pooled all data from the different rigs because separate individual analyses from the individual setups revealed no effects of recording conditions in the different setups.

**Human psychophysics laboratory setup**. We used a similar laboratory setup to our recent experiments[38,64,65]. Briefly, subjects sat in a dark room 57 cm in front of a CRT monitor (85 Hz refresh rate; 41 pixels per deg resolution) spanning 34.1 × 25.6 deg (horizontal x vertical). Head fixation was achieved with a custom head, forehead, and chin rest[64], and we tracked eye movements (of the left eye) at 1 kHz using a video-based eye tracker (EyeLink 1000, SR Research Ltd, Canada). Gray and texture backgrounds (e.g., Figs. 1, 6, 8–10) were always presented at an average luminance of 22.15 cd m$^{-2}$, and the monitor was linearized (8-bit resolution) such that equal luminance increments and decrements were possible around this average for textures and gratings. For the experiments in which we used luminance steps of the background as the visual transients replacing saccade-induced transients (Fig. 5), details of the luminances used are presented below with the experimental procedures.

Human Experiment 1 (Fig. 1) was performed by eight subjects (two female) who were 21–25-year-old. All subjects were naive to the purposes of the experiment, except for subject MB (an author). For Human Experiment 2, the "simulated saccade" version of Human Experiment 1 (Fig. 6), six of the same subjects participated. A control experiment for testing visibility of flashes without saccades and without saccade-like texture displacements (Supplementary Fig. 3a, b) was performed by six of the same subjects plus one non-naive subject, Z.H. (another author).

In the variants of Human Experiments 1 and 2 in which we collected full psychometric curves and perceptual thresholds (e.g., Figs. 2 and 7 and Supplementary Figs. 4 and 7), five subjects (24–29-year-old; one female) participated. Three of these subjects were the same as those who performed Human Experiments 1 and 2 above, confirming that both variants of the experiments (either with a fixed flash contrast or with full threshold calculations) allowed similar conclusions.

In the control experiment (Fig. 5) mimicking the retinal results of Fig. 4d, we collected data from five subjects (25–29-year-old; two female). Two of these subjects were the same as those who performed all experiments.

Human Experiment 3 tested suppression selectivity for low spatial frequencies (Fig. 8). Six subjects (three females, 23–25-year-old) participated, and only subject MB was non-naive. Three subjects had also participated in Human Experiments 1 and 2 and most of their control versions above. A control version of Human Experiment 3 was also performed with black surrounds (Supplementary Fig. 10). This control experiment was performed by the same subjects that participated in Human Experiment 3.

We also ran a variant of Human Experiment 3 describing full psychometric curves of perceptual detectability (Figs. 9 and 10). For each of the real (Fig. 9) or simulated (Fig. 10) variants, we ran four subjects (24–29-year-old; one female; three being the same as those who performed the experiments of Fig. 8).

Across all experiments, we ensured that the same subjects performed real and "simulated" saccade versions of a given paradigm so that we could make meaningful comparisons between these two eye movement conditions.

**Coarse and fine textures**. We created coarse and fine textures (Supplementary Fig. 1a) by convolving a random binary (i.e., white or black) pixel image with a two-dimensional Gaussian blurring filter[66] with the kernel

$$G(x, y) = e^{\frac{-(x^2+y^2)}{2\sigma^2}} \qquad (1)$$

The parameter $\sigma$ of the kernel influenced the amount of blurring. This resulted in textures having effectively low-pass spectral content (Supplementary Fig. 1b) with a cutoff frequency ($f_c$) depending on $\sigma$. As we describe below, we picked cutoff frequencies for coarse and fine textures that resulted in dark and bright image blobs approximating the receptive field sizes of RGCs (for coarse textures) and retinal bipolar cells (for fine textures). In other words, for a given species, coarse textures matched the resolution of RGCs, and fine textures matched the resolution of one processing stage earlier, the retinal bipolar cells.

For the ex vivo experiments with mouse and pig retinae, we assumed receptive field diameters for RGCs of at least 150 $\mu$m (Supplementary Fig. 1c; the parameter $\sigma$ of the Gaussian blurring filter would be half that value), and diameters for bipolar cells of 25 $\mu$m (see Zhang et al.[67]). For human psychophysics experiments, we estimated, from the literature[36], the sizes of human parasol RGC receptive fields at eccentricities >6 deg from the fovea (our flash eccentricities were 7 deg) to be

around 200 $\mu$m. This translated into a cutoff frequency of ~0.68 cycles per deg (cpd) (Supplementary Fig. 1b). Bipolar cell receptive field sizes at this eccentricity were estimated to be 10 $\mu$m (corresponding to a cutoff frequency of ~13.7 cpd), based on sizes of human midget RGC receptive fields in the fovea[36]. When calculating the textures, the actual value of the parameter $\sigma$ (in pixel-dimensions) always incorporated the specific experimental magnification factor between the stimulation screen and the retinal projection of the image. Calculating power spectra for coarse and fine textures confirmed that cutoff frequencies for a given species were consistent with our aimed designs described above (Supplementary Fig. 1b).

For both retinal and perceptual experiments, we normalized pixel intensities in the textures to have uniform variations in luminance around a given mean. In the retinal experiments, we used pixel intensities (from our 8-bit resolution scale) ranging from 0 to 60 around a mean of 30, or ranging from 30 to 90 around a mean of 60 (see Retina electrophysiology experimental procedures below for when each paradigm was used). For the human experiments, textures had a mean luminance of 22.15 cd m$^{-2}$ with undulations in luminance in the texture within the range of 7.5–35.5 cd m$^{-2}$.

As each texture, particularly when coarse, could have patterns of dark and bright blobs that human subjects can remember or interpret as potential shapes/objects/figures, we varied the displayed texture images from trial to trial. This was also necessary to avoid afterimages. We generated sets of 20 coarse and 20 fine textures, which we randomly interleaved across trials. Moreover, the textures themselves were designed to be larger than the viewable display area, allowing us to jitter the displayed sub-rectangle of each texture (within the viewable area of the display) from trial to trial (we jittered the displayed sub-rectangle within a range of 0.6 × 0.6 deg in steps of 0.024 deg). This way, even fine patterns at foveal fixation locations could not be memorized by the subjects across trials.

**Retina electrophysiology experimental procedures**. To simulate saccades in our ex vivo retina electrophysiology experiments, we displaced the texture across the retina in 6 display frames (100 ms at 60 Hz refresh rate). For easier readability, we sometimes refer to these saccade-like texture displacements as "saccades". The textures were displaced in each frame by a constant distance along a linear trajectory. While each "saccade" lasted 100 ms, displacement direction was varied randomly for each "saccade" (uniformly distributed across all possible directions), and "saccade" amplitude could range from 310 to 930 $\mu$m (corresponding to a velocity range of 3100–9300 $\mu$m s$^{-1}$ on the retinal surface). In visual degrees, this corresponds to a velocity range of 100–300 deg s$^{-1}$ and displacement range of 10–30 deg in mice, well in the range of observed mouse saccade amplitudes[68]. In fact, similar to primates, mice also have oculomotor behavior, even under cortical control[69]. For example, they make, on average, 7.5 saccade-like rapid eye movements per minute when their head is fixed[68] (humans make several saccades per second). We used the same retinal displacement range of 310 to 930 $\mu$m for pig retinae. To the best of our knowledge, pig oculomotor behavior has not been documented in the literature. However, with their larger eyeball sizes, our translations of the retinal image would correspond to slower saccades (e.g., small saccades in humans and monkeys), which are also associated with saccadic suppression. Moreover, we showed (Fig. 4) that retinal "saccadic suppression" is not critically dependent on the details of movement kinematics.

Each "trial" consisted of 39 successive sequences that each combined a "saccade" with a probe flash, as follows: there was first a "pre-saccade" fixation of 2 s, then a 100 ms "saccade", followed by "post-saccade" fixation. The background texture was switched on at the beginning of each trial and was translated across the retina during each "saccade". At a certain time from "saccade" onset (delay $d$, range: −177 to 2100 ms), we presented a probe flash. In most cases, the probe flash had a duration of 1 frame (~16 ms). We used 2 frames (~33 ms) in a subset of experiments (mouse: 161 of 688 cells analyzed for "saccadic suppression"; pig: 112 of 228 cells). Results were pooled across these paradigms as they were indistinguishable. For sequences containing no probe flash, the next "saccade" happened 4 s after the previous one. The probe flash was a full-screen positive ("bright") or negative ("dark") stimulus transient. In different experiments, only a subset of possible delays was used within a given set of trials, depending on total recording time for a given retina (see below).

Bright or dark probe flashes could happen in two different ways across our experiments. The results were indistinguishable between the two ways, so we pooled results across them. Briefly, in one manipulation, the probe flash was a homogeneous bright (pixel intensity of 60 in our 8-bit projectors) or dark (pixel intensity of 0) full-screen rectangle replacing the background texture (in these experiments, the textures themselves had intensities ranging from 0 to 60 pixel intensity; see Coarse and fine textures above). This way, the flash contrast from the underlying background luminance was variable (e.g., a bright flash on a bright portion of a texture had lower contrast from the underlying texture than the same flash over a dark portion of the texture). In the second manipulation, the bright and dark flashes were simply luminance increments or decrements (by pixel values of 30 on our 8-bit projectors) over the existing textures (like in our human perceptual experiments). This way, local contrast relationships in the background textures were maintained. In these experiments, the textures themselves had a range of 30–90 pixel intensities and a mean pixel value of 60 (on our 8-bit projectors). Three-hundred thirty-two of 688 cells that we analyzed for "saccadic suppression"

experienced such probe flashes, whereas the rest (356 cells) experienced the homogenous probe flash. For pig retina recordings, we always used the homogenous framework. However, in the subset of pig experiments where the 2-frame probe flash was employed (112 of 228 RGCs), we used a high-contrast probe flash such that a bright flash would be achieved by first going completely dark in the first frame followed by the bright flash in the next frame and vice versa for a dark flash. Again, all data were pooled across these different paradigms because their outcomes were indistinguishable.

The number of trials required during a physiology experiment depended on the number of conditions that we ran on a specific day. For example, testing 7 different flash delays required 15 trials (7 with bright probe flashes, 7 with dark probe flashes, and 1 without probes). In a given experiment, we always interleaved all conditions; that is, in any one of the 15 necessary trials, each of the 39 "saccades" could be followed by a bright or a dark probe at any of the 7 delays, or no probe at all. Moreover, we repeated the total number of conditions (e.g., the interleaved 15 trials) four times per session, and we averaged responses across repetitions. Since one trial typically lasted for 2 min, the example of 15 trials repeated 4 times lasted for ~2 h. This was usually combined with additional conditions (e.g., other background textures), such that typical recordings lasted 10–12 h. If the combination of conditions would have required even longer recordings in a given session, we typically reduced the number of conditions (e.g., we presented flashes at fewer delays).

We sometimes replaced the 100 ms "saccade" with an instantaneous texture jump, to test the sensitivity of retinal "saccadic suppression" (Fig. 3) to the kinematic properties of saccade-like texture displacements (Fig. 4b). Here, the texture simply jumped, in one display frame, from the pre- to the post-displacement position. All other procedures were like described above. Thirty-one RGCs were recorded with this paradigm.

In the control experiments of Fig. 4d, we used no textures at all. The screen was always a homogenous gray field, and the visual event of a "saccade" was replaced by an instantaneous step to a different gray value. The gray backgrounds had intensities between 30 and 90 (on our 8-bit projector). This instantaneous change in intensity caused either a positive contrast step (+0.03 to +0.50 Michelson contrast) or a negative contrast step (−0.03 to −0.50 Michelson contrast). A "trial" consisted of either 57 or 157 successive sequences that each combined a contrast step with a probe flash, as follows: there was first a "pre-step" fixation of 2 s (analogous to "pre-saccade" fixation in texture displacements), then an instantaneous switch to "post-step" fixation. At a certain time from the contrast step (delay: 17, 33, 50, 100, 250, 500, 1000, or 2000 ms), we presented a 2-frame (~33 ms) probe flash. For sequences containing no probe flash, the next contrast step happened 4 s after the previous one. The probe flash was either a uniform negative step of −0.33 Michelson contrast ("dark") or a uniform positive step of +0.33 Michelson contrast ("bright").

Finally, we used other stimuli unrelated to the main experiments to help us characterize RGC types and other receptive field properties (e.g., response polarity, latency, transiency, and spatial receptive fields). These stimuli had the same mean intensities and intensity ranges as the textures used in each experiment. Below, we describe these stimuli for the condition in which the texture intensities ranged from 0 to 60 pixel intensity (represented as grayscale RGB values in the units of our 8-bit projects). In experiments in which the textures ranged in intensity from 30 to 90, all intensities reported below were shifted upward by 30. (1) Full-field contrast steps. ON steps: stepping from 0 to 30 (+1 Michelson contrast) and from 30 to 60 (+0.33) for 2 s. OFF steps: stepping from 60 to 30 (−0.33) and from 30 to 0 (−1) for 2 s. (2) Full-field Gaussian flicker, 1 min. Screen brightness was updated every frame and was drawn from a Gaussian distribution with mean 30 and standard deviation 9. This stimulus was used to calculate the linear receptive field filters of ganglion cells through reverse correlation (spike-triggered averaging of the stimulus history). (3) Binary checkerboard flicker, 10–15 min. The screen was divided into a checkerboard pattern; each checker either covered an area of 55 × 55 μm, 60 × 60 μm, or 65 × 65 μm depending on the recording rig. The intensity of each checker was updated independently from the other checkers and randomly switched between 10 and 50 or 0 and 120. This stimulus also allowed us to calculate the linear filters of cells' receptive fields.

**Human psychophysics experimental procedures**. In Human Experiment 1, we presented a coarse or fine background texture (Fig. 1) for 800–1700 ms in every trial. Over the texture, a white fixation marker (square of 7.3 × 7.3 arcmin) surrounded by a uniform gray circle of 30 min arc radius was presented at one screen location in order to guide gaze fixation onto the marker. The fixation marker was always at 4.8 deg eccentricity from display center, but its specific location was varied from trial to trial (up-right, up-left, down-right, or down-left relative to display center; 45 deg direction from horizontal). After the end of the initial interval, the fixation marker jumped to display center, instructing subjects to generate a saccade.

At a random time from the saccade instruction (47, 94, 153, 200, 247, or 507 ms), a luminance pedestal (probe flash) was applied for one display frame (~12 ms) at one of four locations relative to display center (7 deg above, below, to the right of, or to the left of center). Note that because the display was rasterized (that is, drawn by the computer graphics board from the top left corner in rows of pixels), the actual exact flash time and duration depended on the location of the flash on

the display (but in a manner like other psychophysical experiments studying the same phenomenon, and also in a manner that is unlikely to affect our results). The luminance pedestal consisted of a square of 147.8 × 147.8 min arc in which we added or subtracted a value of 4.8 cd m$^{-2}$ to the texture pattern. Therefore, local contrast within the luminance pedestal was the same as that without the pedestal. Since all of our analyses revealed identical results whether the pedestal was a luminance increment or decrement, we combined these conditions in all analyses. At the end of the trial, subjects had to report their perceived flash location by pressing one of four buttons, corresponding to the four possible flash locations, on a hand-held response box.

As saccadic reaction times were 156.9 ± 3.3 ms s.e.m. across subjects, our choice of flash times above meant that we could analyze trials in which flashes appeared before or after saccade onset, allowing us to obtain full time courses (e.g., Fig. 1). Also, because of the display geometry, the retinal region that experienced a flash before, during, or after a saccade was always a region that was visually stimulated by the texture before flash onset (rather than by the monitor edge or the black surround of the laboratory). Therefore, we maintained pre- and post-flash visual stimulation by texture background, as in the retinal experiments. We also ensured that flash locations were not coincident with saccade goal locations both retinotopically and also in display coordinates. We confirmed in separate analyses that similar effects of suppression (e.g., Fig. 1) occurred for each flash location separately.

We collected 576 trials per session in this experiment. Six subjects participated in six sessions each, and the remaining two participated in three or four sessions.

Human Experiment 2 (Fig. 6) was identical, except that the initial fixation marker was presented at display center and remained there for the entire duration of a trial. Instead of instructing a saccade 800–1700 ms after fixation marker onset, we translated the entire background texture (switched on at trial onset) rapidly to simulate a saccade-like image displacement. Texture displacement consisted of a 6-frame translation at a speed of 176 deg s$^{-1}$. Note that, because of our display refresh rate and geometry, this meant a slightly larger displacement (of 12.4 deg) when compared to the saccade sizes in Human Experiment 1. However, we chose this translation because it resulted in a sufficiently fast average speed of the displacement (average speed in the real saccades of Human Experiment 1 was 160 deg s$^{-1}$). This choice is not problematic because our retinal experiments revealed that visual mechanisms related to saccadic suppression were not sensitive to parameters of individual motion patterns (Fig. 4b).

In this experiment, the texture displacement happened in a diagonal direction to simulate the directions of saccadic displacements of Human Experiment 1 (and also to dissociate the direction of motion flow from the locations of the flashes, again as in Human Experiment 1). For example, the texture could move globally down-right, as might be expected (in terms of image motion) if subjects made upward-leftward saccades in Human Experiment 1. Also, flash times were chosen relative to the onset of texture displacement from among the following values: −35, −24, 24, 47, 84, 108, 141, 200, 259, 494 ms.

All subjects participated in ten sessions each in this experiment.

We also performed a control experiment, in which there was neither a real saccade (Human Experiment 1) nor a texture displacement (Human Experiment 2), but otherwise identical to these two experiments. Subjects simply fixated display center, and we presented (after 1200 to 2400 ms from trial onset) a luminance pedestal exactly as in Human Experiments 1 and 2. To obtain full psychometric curves, we varied the luminance increment from among six values (Supplementary Fig. 3a, b). Subjects performed two sessions each of this experiment (600 trials per session).

To explore perceptual thresholds in a more quantitative manner for Human Experiments 1 and 2, we also performed additional real or simulated saccade experiments collecting full psychometric curves (Figs. 2 and 7; and Supplementary Figs. 4 and 7). The logic of both additional experiments (real or simulated) was the same as that of Human Experiments 1 and 2, except that we varied the luminance of the probe flash from trial to trial (like in the above control experiment of flash visibility; Supplementary Fig. 3a, b). As this endeavor (allowing us to measure full psychometric curves) was very data intensive, we reduced the time samples relative to saccade onset or texture displacement onset at which we probed perceptual performance. For the experiment with real saccades, we used an automatic procedure to detect saccade onset in real-time based on eye velocity, as described by Chen and Hafed[39]. We then presented the probe flash at 42, 65, 88, or 148 ms after saccade detection. These times were chosen because they covered intervals of maximum perceptual saccadic suppression as well as recovery, allowing us to get a time course of perceptual threshold elevation associated with saccadic suppression. In subsequent data analyses, we confirmed that these flash times were as planned (within the expected variability due to the asynchronous nature of saccade times relative to display update times; Fig. 2). For the experiment with simulated saccades, we presented the probe flash at −24, −12, 48, or 96 ms relative to the onset time of the texture displacement. In this case, we introduced a new negative time sample to the set (−12 ms) because the original Human Experiment 2 did not probe this particular time (e.g., Fig. 6). It was therefore important to clarify that the time course of perceptual suppression for simulated saccades was continuous and well-behaved, exactly like that for real saccades.

In order to also estimate perceptual thresholds online in these additional experiments, and therefore optimize the numbers of trials needed, we applied an adaptive QUEST procedure[40] on each randomly interleaved condition. Specifically,

the first 40 trials of each randomly interleaved condition (e.g., flash time –24 ms and coarse texture, or flash –12 ms time and fine texture, and so on) were part of the QUEST procedure. The remaining trials in the session interleaved four additional flash luminances per condition, which were chosen to lie around the threshold luminance of each condition as detected by the QUEST procedure. These additional flashes had luminances that were ±1 or ±2 times a pre-defined luminance increment for a given condition, depending on the detected threshold and earlier pilot data. Specifically, if the detected threshold (according to QUEST) was very low (e.g., no suppression effect), the pre-defined luminance increment was 1 step of luminance (dictated by the luminance resolution of our display; Supplementary Fig. 3a). That is, the four additional flashes were at ±1 and ±2 display-determined luminance steps from the detected threshold. If the detected threshold (according to QUEST) was high (e.g., strong suppression), we made the pre-defined luminance increment 2 or 5 display-determined luminance steps (that is, ±2 and ±4 display-determined luminance steps or ±5 and ±10 display-determined luminance steps, respectively). This allowed fitting the psychometric curves during subsequent data analyses, including measurements from the full dynamic range of perceptual performance. The reasoning behind this approach is as follows: depending on the amount of perceptual saccadic suppression to be expected per condition (e.g., peak suppression during saccades or texture displacements, or very weak suppression during recovery), it is expected that the psychometric curves would be shifted by different amounts from baseline depending on the particular condition (e.g., flash time or coarse versus fine texture). Finally, also note that we only used bright flashes in these particular experiments instead of both bright and dark flashes. In total, we collected 240 trials per condition per subject.

In yet another control experiment for Human Experiments 1 and 2, we mimicked the retinal results of Fig. 4d. Subjects fixated a central fixation spot over a gray background. The background had one of eight luminances (22.4, 30.24, 38.08, 45.92, 53.76, 61.6, 69.44, 77.28 cd m$^{-2}$). After a random initial fixation duration (similar to Human Experiment 2), the luminance of the background was changed suddenly (in one display frame update) to one of the remaining seven luminances. This meant that across trials, we had seven total levels of contrast change in the background as our visual transient. At one of five different possible times relative to the time of background luminance change (–24, –12, 36, 72, or 108 ms), a luminance pedestal was flashed briefly, exactly like in Human Experiments 1 and 2. We ensured that the contrast of the flash (relative to the currently displayed background luminance) was always the same across all trials. We also ensured that baseline visibility of the pedestal in the absence of the contrast change was at ceiling performance (see the longest sampled time value in Fig. 5, demonstrating near perfect detection performance for all background luminance steps). Subjects maintained fixation throughout all trials and simply reported the locations of the brief flashes. Subjects performed one session, each, of this experiment with 1120 trials per session.

In Human Experiment 3 (Fig. 8), the flashes of Human Experiments 1 and 2 were replaced by vertical Gabor gratings having one of five different spatial frequencies (0.41, 0.85, 1.71, 3.42, 4.56, or 6.8 cpd). The contrast of the grating (defined as the difference between maximum and minimum luminance in the grating divided by the sum of the same luminances) was 14.3%. Spatial phase was randomized from trial to trial, and the $\sigma$ parameter of the Gaussian envelope was 0.49 deg. Also, a virtual monitor of 20 deg diameter was present at display center at the time of Gabor grating flashes. The virtual monitor had a uniform gray luminance equal to the average of the textures used in Human Experiments 1 and 2. Surrounding the virtual monitor, a coarse or fine texture could be visible.

In one block of trials, subjects generated saccades towards display center using the same procedures as in Human Experiment 1. Grating flash times were similar to Human Experiment 1, and the subjects performed 6 sessions each (576 trials per session).

In another block of trials, subjects maintained fixation at display center. In one third of the trials, the virtual monitor and surrounding texture did not move. These trials provided us with "baseline" visual performance (i.e., without saccades or virtual monitor displacements). It was necessary to have these trials because perceptual visibility of different spatial frequencies is not equal due to the well-known human contrast sensitivity function[70]. Therefore, we needed to establish "baseline" grating visibility first and then compare the effects of saccades or saccade-like virtual monitor displacements on such visibility. In the remaining two thirds of the trials, the virtual monitor and surrounding texture initially appeared displaced from display center at a location near one corner of the display and along one of the diagonal directions. After 800–1700 ms, the virtual monitor and surrounding texture were translated rapidly towards display center to simulate visual flow associated with the diagonal saccades of the real-saccade version of the paradigm (the translation parameters were similar to Human Experiment 2). Grating flashes happened 84 ms or 108 ms after virtual monitor and texture displacement. Note that we reduced the number of flash times here because of the larger number of conditions (five different spatial frequencies of the Gabor gratings) that needed to be collected. However, our data were consistent with all other experiments in terms of recovery time courses of suppression (e.g., Figs. 1, 6, 8 and Supplementary Figs. 8–10).

As the initial displaced position of the virtual monitor (and texture) provided a cue to subjects that grating onset was expected soon, and because such a cue was not present in the one third of trials without image motion, we equalized subject

expectations across these conditions by dimming the fixation point to black from the time of image motion onset until 200 ms after flash onset (equal timing was ensured in the one third of trials without image motions, such that the same expectation of grating onset was established by fixation marker dimming). The fixation marker then disappeared, and subjects had to report flash location.

Subjects performed six sessions each of this condition, with 576 trials per session (two subjects performed seven and five sessions each instead of six).

We also repeated the same experiment but with a black surround around the virtual monitor instead of a coarse or fine texture. Note that a black surround is theoretically equivalent to an infinitely coarse surround. We therefore expected results conceptually similar to those with a coarse surround. Also, in this control experiment, we randomly interleaved all trial types together in the same session (fixation with virtual monitor displacement, real saccade, and fixation with neither virtual monitor displacement nor saccade). This allowed us to further confirm that our results from Human Experiment 3 were not influenced by the separate blocking of real saccade trials and virtual monitor displacement trials.

We also repeated Human Experiment 3 to collect full psychometric curves, like we did for Human Experiments 1 and 2 above. In these additional experiments, because of the data-intensive nature of full psychometric curves, we concentrated on the three lowest spatial frequencies of the Gabor gratings. This was sufficient to observe selectivity or lack of selectivity of perceptual suppression as a function of spatial frequency (e.g., Fig. 8). More importantly, these three lowest spatial frequencies were associated with ceiling baseline visibility (Fig. 8), thus simplifying interpretations of any suppression that we would observe. The experiments were the same as Human Experiment 3, except that the contrast of the flashed Gabor grating was varied from trial to trial. We used a similar adaptive procedure to that used in Figs. 2 and 7 to select contrast from trial to trial, in order to optimize finding perceptual thresholds and fitting of psychometric curves (see procedures above). We also used the same online saccade detection algorithm as in the experiments of Fig. 2 to decide on the time of Gabor grating flash onset (see procedures above). For both real and simulated saccade variants of these experiments, we used two times relative to the "saccade" event, one within a period associated with strong perceptual suppression and one at a late time point associated with perceptual recovery (see Figs. 9 and 10).

**Retina electrophysiology data analysis and statistics**. Low-density MEA recordings were high-pass filtered at a 500 Hz cutoff frequency using a tenth-order Butterworth filter. We extracted spike waveforms and times using thresholding, and we semi-manually sorted spikes using custom software. For high-density MEA recordings, we performed spike sorting by an offline automatic algorithm[71] and assessed the sorted units using UnitBrowser[72]. We judged the quality of all units using inter-spike intervals and spike shape variation. Low-quality units, such as ones with high inter-spike intervals, missing spikes, or contamination, were discarded. All firing rate analyses were based on spike times of individual units.

We first characterized the properties of RGCs. We calculated linear filters in response to full-field Gaussian flicker and binary checkerboard flicker by summing the 500-ms stimulus history before each spike. The linear filters allowed determining cell polarity. Specifically, the amplitude of the first peak of the filter was determined. If the peak was positively deflected, the cell was categorized as an ON cell; if negatively deflected, the cell was an OFF cell. ON cells were later always analyzed with respect to their responses to bright probe flashes in the main experiment, and OFF cells were analyzed with dark probe flashes. We determined the spatial receptive fields of RGCs by calculating the linear filters for each region (checker) defined by the binary checkerboard flickering stimulus. The modulation strength of each linear filter, measured as the s.d. along the 500 ms temporal kernel, is an estimate for how strongly that region drives ganglion cell responses. We fitted the resulting 2D-map of s.d. values with a two-dimensional Gaussian and took the 2-$\sigma$ ellipse (long axis) as the receptive field diameter. For all other figures and analyses, we converted spike times to estimates of firing rate by convolving these times with a Gaussian of $\sigma = 10$ ms standard deviation and amplitude 0.25 $\sigma^{-1}e^{1/2}$.

For each RGC, we used responses to full-field contrast steps to calculate an ON-OFF index, a transiency index, and a response latency index. These indices were used to characterize the properties of RGCs (Supplementary Fig. 6) that we included in our analyses. The ON-OFF index was calculated by dividing the difference between ON and OFF step peak response by their sum. The resulting index values ranged between –1 (OFF) and +1 (ON) and were then scaled to span between 0 (OFF) and +1 (ON). The transiency index was defined as the ratio of the response area within the first 400 ms and the total response area spanning 2000 ms. The resulting index had a value of 1 for pure transient cells. Response latency was calculated as the time from stimulus onset to 90% of peak response. This value was normalized to the maximum response latency in our dataset to create the response latency index.

To quantify retinal "saccadic suppression", we first determined a "baseline response", defined as the response to a probe flash ~2 s after texture displacement onset (delay between 1967 and 2100 ms, depending on the specific flash times used in a specific experiment). This baseline response was compared to responses of the same cell to the same flash when it occurred at an earlier time (i.e., closer in time to the "saccade"). Usually, the saccade-like texture displacements themselves caused significant neural responses even without flashes ("saccade-response", e.g., Fig. 3b), and the responses to the flashes were superimposed on these "saccade-responses"

(Fig. 3c). We therefore first isolated the component of the responses caused by the flashes by subtracting the "saccade-responses" from the composite responses.

To get a robust estimate of the response to "saccades" alone (i.e., without any flashes), we averaged spike rate from before "saccade" onset up until the next "saccade" onset for conditions in which no flash was presented, or until just before the flash onset for conditions in which a "post-saccade" flash was presented. This was done for each of the 39 successive "saccades" in a given trial.

We then computed a neural modulation index, ranging from −1 to +1. A value of −1 represents complete suppression of flash-induced responses, whereas +1 indicates "complete enhancement" of flash-induced responses (that is, there was only a response to a flash after saccades, but not to a flash in isolation). A modulation index of 0 meant no change in flash-induced response relative to the "baseline" response. The modulation index of an RGC for a given flash delay $d$ after "saccade" onset was calculated as $(r_d - r_b)/(r_d + r_b)$ where $r_d$ is the peak firing rate for the flash-component of the response (see above for how we isolated this from the composite "saccade" +flash response) and $r_b$ is the peak firing rate for the baseline flash response (i.e., the same flash but occurring ~2 s away from any "saccade"; see above). In all cases, peak firing rate was estimated after averaging responses from all repetitions of a given condition (delay d or baseline) for a given RGC. For ON cells, the modulation index was based only on responses to bright flashes, and for OFF cells, it was based on responses to dark flashes. For some analyses, we also calculated modulation indices of RGCs for each of the 39 individual "saccades" using the same procedure.

In some cells and trials, individual "saccades" from the sequence of 39 were discarded. This happened when the baseline response peak was <60% of the median baseline response peak across the 39 "saccades" of a given trial. We did this to ensure that our modulation indices were not marred by a numerator and denominator approaching zero (e.g., if both flash and baseline responses were weak). We did, however, re-include sequences in which the peak response to the flash after the "saccade" was above the median baseline response peak (across the 39 "saccades"). This was done in order to re-include sequences (if discarded by the first step) for which the baseline flash response was weak but a flash after "saccades" nonetheless gave a robust response. For example, this could happen if a cell did not respond to a flash in isolation but the "saccade" enhanced the response to a flash following it. Our main results (e.g., Fig. 3) were highly robust to such scenarios.

Finally, to perform statistics, we applied tests at either the individual cell level or at the level of the population. At the individual cell level, we determined whether a given RGC's modulation index for a probe flash presented at a given delay was significantly different from 0 (i.e., "Is the response of this cell modulated by the "saccade"?"). For this, we performed a one-tailed sign test of the null hypothesis that the 39 individual modulation indices came from a distribution with zero median against the alternative hypothesis that the median was below (for negative modulation index) or above (for positive modulation index) zero. The modulation index was considered significant (i.e., the flash response was modulated by the "saccade") at $p < 0.05$ if the test had a power $(1 - \beta)$ of at least 0.8. At the population level, we determined whether the retinal output as a whole was modulated by "saccades". For this, we performed a two-tailed Wilcoxon signed-rank test of the null hypothesis that the median of the distribution of modulation indices did not differ from 0. Lastly, we tested whether the modulation index of the population was significantly different across textures. For this, we performed a two-tailed Wilcoxon signed rank test of the null hypothesis that the median of the distribution of modulation indices did not differ across textures. Since our modulation index was based on responses to the brief probe flashes, it could only be computed for cells that did respond to these flash stimuli (mouse: $N = 688$ of 1423 recorded cells; pig: $N = 228$ of 394). Only these cells, showing a measurable baseline flash response, were included in our analyses for retinal "saccadic suppression" (Fig. 3e and Supplementary Fig. 5).

To quantify retinal "saccadic suppression" in our control experiments with structure-free uniform backgrounds and luminance steps in place of textures and texture displacements (Fig. 4d), we used the same analyses and statistical procedures to those described above for the texture displacement paradigm. The only difference was that instead of 39 successive "saccades" in a trial, we now had either 57 or 157 successive full-field luminance steps (depending on experiment setting). Twenty-two of 57 or 66 of 157 steps had a Michelson contrast in the range of ±0.03 to ±0.15 and these steps were used to quantify suppression for low-contrast luminance steps. Twenty-four of 57 or 58 of 157 steps had a Michelson contrast in the range of ±0.20 to ±0.40 and were used to quantify suppression for high-contrast luminance steps. From the perspective of visual transients across the retina, low-contrast luminance steps are equivalent to fine texture displacements over receptive fields, and high-contrast luminance steps are equivalent to coarse texture displacements. This is simply because of the spatial relationship between receptive field sizes and texture spatial scales: a fine texture presents both dark and bright blobs within individual receptive fields before and after the texture displacement (resulting in a low-contrast change in luminance over the receptive fields); on the other hand, a coarse texture has dark or bright blobs that are of similar size to the receptive fields (resulting in the potential for a very large contrast change in luminance over the receptive fields after the texture displacement). As shown in Fig. 4d, low and high-contrast luminance steps resulted in the modulation of ganglion cell responses to the probe flashes that was reminiscent of the modulation observed after displacement of fine and coarse textures, respectively

(also validated perceptually in Fig. 5). Similar to the texture displacement paradigm, the modulation index was based on responses to brief probe flashes, and it could therefore only be computed for cells that did respond to these flash stimuli ($N = 376$ of 650 recorded RGCs in mouse). The modulation index for ON RGCs was calculated from responses to bright probe flashes, and that for OFF RGCs was calculated from responses to dark flashes.

**Human psychophysics data analysis and statistics.** We analyzed eye movements in all trials. We detected saccades using established methods[39,73], and we manually inspected all trials to correct for mis-detections. In experiments requiring a saccade (e.g., Fig. 1), we excluded from analysis any trials with premature (before saccade instruction) or late (>500 ms reaction time) saccades. We also rejected all trials in which saccades landed >0.5 deg from the saccade target. In experiments requiring fixation, we excluded from analysis any trials in which a saccade or microsaccade happened anywhere in the interval from 200 ms before to 50 ms after any flash or grating onset.

For experiments with saccades (e.g., Fig. 1), we obtained time courses of perception by calculating, for each trial, the time of flash or grating onset from saccade onset. We then binned these times into 50 ms bins that were moved in 5 ms bin-steps relative to saccade onset. Within each bin, we calculated the proportion of correct trials, and we obtained full time courses of this perceptual measure. We obtained time course curves for each subject individually, and we then averaged the curves for the individual subjects in summary figures. All of our analyses were robust at the individual subject level as well (e.g., Supplementary Fig. 2).

For experiments with simulated saccades (i.e., saccade-like texture displacements), or background luminance steps (Fig. 5), there were discrete flash or grating times relative to "simulated saccade" onset, so no temporal binning was needed. At each flash or grating time, we simply calculated the proportion of correct trials.

When we fitted performance to psychometric curves (e.g., Supplementary Fig. 3a, b), we used the psignifit 4 toolbox[74], and we used an underlying beta-binomial model. In all psychometric curve fits, we also included lapse parameters among the fitted parameters, in order to account for potential small deviations from either perfect ceiling performance or perfect floor (chance) performance at the extremes of the psychometric curves.

We also used the same toolbox to analyze the variants of Human Experiments 1 and 2 in which we collected full psychometric curves (Figs. 2 and 7). For these experiments, we defined the threshold of an individual subject as the flash luminance level that resulted in correct perceptual performance at a value of 62.5% of the total dynamic range of the subject's psychometric curve (that is, 62.5% of the dynamic range of the fitted psychometric curve after the inclusion of lapse rates). We then plotted the value of such threshold as a function of flash time relative to real or simulated saccade time.

For some analyses of Human Experiment 3 and its control version, we calculated a "suppression ratio" as a visualization aid (e.g., Fig. 8). This was obtained as follows. For a given spatial frequency grating, we calculated the fraction of correct trials within a given time window (from either simulated or real saccade onset) divided by the fraction of correct trials for the same spatial frequency when there was neither a saccade nor a virtual monitor and texture displacement (i.e., baseline perception of a given spatial frequency). This ratio therefore revealed the effect of suppression independently from the underlying visibility of any given spatial frequency[14]. However, note that we also report raw proportions of correct trials in all conditions.

All error bars that we show denote s.e.m. across individual subjects, except where we report individual subject analyses and control analyses. For individual subject performance, error bars denote s.e.m. across trials; for control analyses, error bars denote 95% confidence intervals (e.g., Supplementary Fig. 3a, b) or s.d. (e.g., Supplementary Fig. 3d, f). All error bar definitions are specified in the corresponding figures and/or legends.

To statistically validate if the time courses for perceptual localization performance for saccades across the different background textures (coarse versus fine) differed significantly from each other (e.g., Fig. 1), we used a random permutation test with correction for time clusters of adjoining significant $p$-values[37,38]. First, for each time bin, we calculated a test statistic comparing performance for coarse versus fine background textures. This test statistic was the difference between the proportion of correct responses for the different textures. Then, we performed a random permutation with 1000 repetitions for each time bin; that is, we collected all trials of both conditions, within a given time bin, into a single large set, and we randomly assigned measurements as coming from either coarse or fine textures, while at the same time maintaining the relative numbers of observations per time bin for each texture condition. From this resampled data, we calculated the test statistic again, and we repeated this procedure 1000 times. Second, we checked, for each time bin, whether our original test statistic was bigger than 95% of the resampled test statistics (i.e., significant), and we counted the number of adjoining time bins that were significant at this level (i.e. clusters of time bins in which there was a difference between coarse and fine textures). We then repeated this for all 1000 resampled test statistics. The $p$-value for our original clusters was then calculated as the number of resampled clusters that were bigger or the same size as the original clusters, divided by the total number of repetitions

(1000). This procedure was described in detail elsewhere[38]. We followed a conservative approach, paying no attention to which bins in the resampled data formed a cluster of time bins. As discussed elsewhere[38], our statistical analysis constituted a highly conservative approach to establishing significance of differences between time courses for coarse and fine textures. In Human Experiment 3, we used the same approach to compare time courses of suppression ratio for coarse and fine surround contexts with real saccades.

For Human Experiment 2, we had discrete flash times relative to texture displacement onset. Here, the comparison between coarse and fine textures was tested with a Bonferroni-corrected $\chi^2$ test at corresponding flash times. To compare between real and simulated saccades in Human Experiments 1 and 2, we also ran a Bonferroni-corrected $\chi^2$ test. We only considered time bins in the real saccade data that corresponded to the discrete flash times in the simulated saccade data. A Bonferroni correction was necessary because we tested the same data sets on multiple time bins with the same hypothesis (that there is a difference in time courses).

In Human Experiment 3, we also compared suppression ratios for real and simulated saccades for a given texture surround. We again used a Bonferroni-corrected $\chi^2$ test. This was justified because within a given surround, baseline data were the same for real and simulated saccades. Therefore, the relationship between the proportion of correct localizations and suppression ratio was identical. In contrast, testing suppression ratios between fine and coarse surrounds in the same experiment with a $\chi^2$ test was not applicable because baseline values differed. Therefore, we used instead a random permutation test with 5000 repetitions. To compare the different spatial frequency Gabor gratings in one bin or time stamp, we used the Kruskal–Wallis test.

For the psychometric versions of Human Experiment 3 (Figs. 9 and 10), we used similar analyses on perceptual thresholds to those used in the psychometric versions of Human Experiments 1 and 2 (Figs. 2 and 7).

All analyses were done in MATLAB (The MathWorks Inc).

**Reporting summary**. Further information on research design is available in the Nature Research Reporting Summary linked to this article.

## Data availability

All data presented in this paper are stored and archived on secure institute computers and are available upon reasonable request.

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

## Acknowledgements

Andreas Hierlemann provided the HiDens CMOS MEA system and helped establish our high-density MEA recordings. Roland Diggelmann helped in setting up the pipeline (including providing code) for automatic spike sorting of high-density MEA recordings. Our work was supported by funds of the Deutsche Forschungsgemeinschaft (DFG) to the Werner Reichardt Center for Integrative Neuroscience (EXC 307) and to T.A.M. (MU3792/3-1). T.A.M. received support from the Tistou and Charlotte Kerstan Foundation. T.A.M. and Z.M.H. were also supported by an intra-mural funding program (Projekt 2013-05) of the Werner Reichardt Center for Integrative Neuroscience. F.F. was supported by a Swiss National Science Foundation Ambizione grant (PZ00P3_167989), and M.P.B. and Z.M.H. were further funded by the SFB 1233 on "Robust Vision" (DFG) - project number 276693517.

## Author contributions

S.I., M.B., T.A.M., Z.M.H. designed the overall study; S.I., M.B., T.A.M., Z.M.H. designed experiments; S.I. performed ex vivo retina experiments; M.B., Z.M.H. performed human psychophysics experiments; S.I., M.B., F.F., T.A.M., Z.M.H. analyzed data; S.I., M.B., F.F., T.A.M., Z.M.H. wrote manuscript.

## Competing interests

The authors declare no competing interests.
