## [Peer Review File · Nature Communications]

Reviewers' comments:

Reviewer #1 (Remarks to the Author):

In this manuscript, Idrees and colleagues report a series of experiments addressing a contentious question - the mechanisms of saccadic suppression. In human subjects, they find that: (1) detection of luminance increments on low-pass filtered noise textures is impaired around the time of saccades, particularly when a broad blurring filter is used ('coarse'); (2) impairment also occurs when around the time of saccade-like image translations, although the effects are more prolonged; and (3) losses in visibility of grating patches with both real and simulated saccades are spatial frequency selective when surrounded by a coarse (i.e. highly blurred) texture, but spatial frequency invariant when surrounded by a fine (less blurred) texture. In ex vivo retina electrophysiological recordings, they find suppression of responses to luminance increments around the time of simulated saccades, again particularly when presented on highly blurred, 'coarse' textures. The authors conclude that image processing in the retina can account for a large component of saccadic suppression.

Overall, I have mixed feelings regarding this submission. The behavioral experiments comparing real and simulated saccades reveal some potentially interesting results, but I have concerns regarding the methods used to quantify suppression. The electrophysiological result is intriguing and potentially very important. However, it is not very well connected to the third set of behavioral experiments. I've elaborated on these points below.

I have reservations about the use of percent correct detection of a fixed stimulus (luminance increment or grating contrast) to quantify visual sensitivity. For example, the results shown in Figure 1D are interpreted as evidence that saccadic suppression with coarse textures starts earlier and recovers later than with fine textures. However, the floor/ ceiling limits at chance/perfect performance make it very difficult to distinguish a change in the time course of suppression from a change in magnitude.

These problems are compounded when the magnitude of suppression is expressed as the ratio of percent correct performance relative to baseline (no saccade) some of which are at ceiling or floor (Figure 4). As a result, the spatial frequency tuning profiles reported may not be robust. To exemplify the problem, consider the simple case where suppression shifts an underlying psychometric function rightward along the contrast axis. Calculation of a suppression ratio in this manner will return different values at different contrast levels. While I appreciate that it requires a significant investment of time, the only real solution here is to manipulate luminance/contrast and measure thresholds.

It should be acknowledged that this is not the first study to compare the effects of real and simulated saccades and that some of the findings have been previously demonstrated. In particular, Diamond et al (2000) also reported suppression of targets presented on a patterned background with simulated saccades, that was more prolonged than with actual saccades.

The demonstration that retinal ganglion cell responses to a target flash are suppressed around the time of simulated saccades might be the most interesting finding reported in the manuscript. Given the title, I was expecting the authors to go on and provide some insight into how processing in the retina gives rise to this suppression. However, this isn't really developed in the current manuscript. It is also unclear whether any of the results obtained with grating stimuli presented on gray backgrounds could be accommodated by retinal processing. The end result is a manuscript that, despite its substance, feels somewhat disjointed and incomplete.

Minor:

Line 65, note that given a single frame presentation, the actual duration of the localised probe flash would be significantly less than 12ms.

Line 77. The highlighted time intervals in Figure 1 were difficult to see in my version.

Line 168-169. Some referencing inconsistencies here.

Reviewer #2 (Remarks to the Author):

Hafed, Idrees and colleagues have measured in isolated retinæ of mice and pigs the effect of image motion on responsiveness of retinal ganglion cells. They find properties of "saccadic suppression" similar to those measured in a parallel, behavioral study in human participants. This "saccadic suppression" induced by image motion outlasted suppression induced by saccades. The authors argue to have identified a far-reaching contribution of visual processing mechanisms to saccadic suppression at the level of the retina, which would be in place without motor-based suppression commands, i.e. efference copy or corollary discharge.

This is an exciting study including heroic and state-of-the-art experiments on isolated retinæ. Data analysis is fully adequate, results are clear. The fact that neural correlates of effects, which have been described behaviorally here or before, can potentially be found already at the level of the retina, is really exciting. Yet, I do have major concerns about the experimental approach and the interpretation of data, and the conclusions drawn from these results. I detail my concerns below:

Major:

1) The experimental apparatus: if I understood it correctly, the stimulus projector was running at 60 Hz (page 32). To my best understanding, this is not adequate. I have to say that I am not an expert of mouse or pig retina, but is it clear that 60 Hz is way above flicker fusion frequencies of retinal cells in these species? If not, I would have a major concern about intra-retinal signal processing during stimulation. In addition: I guess that the presentation of a textured pattern moving at a hundred degrees per second is not easy if not impossible at that low frame rate. The authors must comment on that.

2) "Saccadic suppression" in retina, Figure 1: The authors have determined the response for a "probe flash" after presentation of a "saccade like texture displacement". My first question is: was the texture visible at all times, and only moved at $t=0$ ms. Or was the texture switched on and started moving simultaneously at $t=0$ ms, and switched off after movement. This is critical for the following. Importantly, this texture displacement (or texture onset plus movement) induces retinal activity, as shown in Figure 2B. Hence, the probe-flash can be considered a second stimulus. Previous work (e.g. Mayo and Sommer, *Neuronal Adaptation Caused by Sequential Visual Stimulation in the Frontal Eye Field*, *J.Neurophysiol.*, 2008) has shown, that at some stages of visual processing, the response to this second stimulus can be clearly reduced if it is presented shortly after the first stimulus. Actually, Mayo's and Sommer's Figure 4 is very similar to Figure 2E and 2F of this manuscript. Yet, Mayo and Sommer consider response attenuation for early second stimuli to be an effect of adaptation. Hence, from my point of view, a critical control experiment is missing here. In this control experiment two probe-flashes should be presented, with Inter-Stimulus-Intervals (ISIs) as used in the study for texture and probe. Only, if in such case (i.e. two probe-flashes with different ISIs) the second response would be identical for all ISIs, one can conclude that the movement of the texture caused something like saccade-like suppression. If, however, the response amplitude for the second probe-flash would be very much reduced, then the effect as shown in Figure 2 should be considered an inhibition due to stimulus-repetition rather than "saccadic suppression".

3) Figure 3: It is already known that the presentation of a textured pattern at high speed induces effects similar to saccadic suppression. This has been shown, e.g. by Diamond, Morrone and colleagues (*J.Neurosci.*, 2000), in their Figure 5. Nevertheless, I agree with the authors that it is exciting to have identified a neural correlate of this behavior at the level of the retina.

4) Figure 3, continued: also the finding that a saccade-like texture displacement induces effects similar to saccadic suppression, which last longer than saccadic suppression induced by real eye

movements, is already known. As an example, see Figure 5 of Diamond et al., *J. Neurosci.*, 2000. These authors mentioned "More remarkable is that the recovery of sensitivity is slower for mirror motion than for saccades" (citation from their paper). So, the conclusion drawn by Diamond and colleagues matches the idea of the authors of the current study.

Minor:

- 1) Thiele et al., *Science*, 2002, should also be cited when referring to previous studies on saccadic suppression (page 1 and 2)
- 2) Even more important would be to mention already early on and explicitly "masking" as alternative basis of saccadic suppression compared to mechanisms like efference copy or corollary discharge, i.e. in the first paragraph of page 2. I suggest to cite work from Wurtz and colleagues, e.g. Campbell & Wurtz, 1978, but also the excellent review: Wurtz, *Vis Res*, 2008.
- 3) Lines 112, 113: given my concerns above, sentences like "Rather, suppression was a result of specific spatio-temporal retinal-circuit image processing initiated by the texture displacements" might not be true.
- 4) Line 126: like myself, many readers might not be aware of the native oculomotor behavior of mice and pigs. Authors should briefly explain how often they make saccades, at what max speed, etc.
- 5) When referring to Castet and colleagues concerning intrasaccadic perception of visual motion, I suggest to refer also to an earlier work on the same issue: Ilg and Hoffmann, *Vis Res*, 1993.
- 6) The authors should check their reference list. As an example, a Castet et al. paper is listed twice.

Reviewer #3 (Remarks to the Author):

The detection of a briefly presented visual stimulus is impaired shortly before and during a saccade. The search for the mechanisms of this saccadic suppression is as old as the discovery of the phenomenon. Broadly speaking, two classes of explanations have been advanced since the 1970s. First, a set of visual mechanisms mostly based on forward and backward masking. Second, a central process based on suppressive influence of a corollary discharge. Although the discussion about the two classes of explanation is still ongoing I believe it is fair to say that a contribution of visual masking is undoubted in suitable conditions. The question, then, is how much of saccadic suppression can be explained by visual processes and whether a central process is needed.

The present study makes two contributions to the discussion. First, it compares the influence of texture patterns on saccadic suppression and masking in humans. Second, it records retinal responses to saccade-like motion of texture patterns and flashes in mouse and pig retina. In comparing the findings from these experiments it shows some strong similarities between saccadic suppression and visual masking but also some clear differences.

In the human experiments, participants made saccades across a background of fine or coarse texture while a brief luminance pedestal was presented around the onset of the saccade. The results (Fig. 1) show that suppression begins earlier and lasts longer for the coarse than for the fine texture background. Peak suppression, on the other hand, was not different. Fig. 3 shows the results of a similar experiment with simulated saccades, i.e., a saccade-like displacement of the texture pattern. In this case, the onset of suppression (at least for the coarse pattern) is even earlier and suppression lasts even longer than for the real saccades in Fig. 1. This is clear indication of strong masking in these conditions, and of a difference in masking between the coarse and the fine textures. Unfortunately, there is no data presented for the interesting time shortly before and at motion onset, which is the time of peak suppression. Thus, it is unclear whether real and simulated saccades differ at that time. Yet, the longer time course of masking suggests that suppression is somewhat different from masking.

Fig. 4 shows a comparison of real saccades and saccade-like image motion for masking by a

surrounding pattern. Unlike in Experiment 1, the suppression during saccades is much stronger than the suppression produced by texture displacement (cf Fig. 4B vs 4D and 4E vs 4F). Therefore, this experiment would seem to show some similarity between masking and suppression but also that suppression requires more than just masking.

The experiments on mouse and pig retina measure the influence of saccade-like retinal image motion on RGC responses to brief flashes. Similar to the human suppression/masking data coarse texture patterns produce more masking than fine texture patterns. This would suggest that part of the masking properties originate from the retina. However, the time course of the retinal effects is vastly different from that of saccadic suppression in humans. The retinal effects last for several hundred milliseconds while the suppression during real saccades in humans last only for about 50-100ms after saccade onset. Moreover, the retinal effects seem to produce only forward masking, i.e. the reduction of responses to flashes presented after motion onset. Responses to flashes presented before motion onset, i.e. in a backward masking situation, appear only weakly modulated (Fig. 2E). Like in the human data of Fig. 3 there are not data points at or near motion onset so it is difficult to compare to the most interesting time range for saccadic suppression, i.e., slightly before and at saccade onset.

In summary, the paper makes some interesting contributions on visual masking, its origin in the retina, and its contribution to the impairment of visual detection during a saccade. Since the data from before and near saccade/motion onset is either missing or inconsistent the conclusion that explicit motor-based suppression is not needed is too strong. Since in some conditions visual masking is longer lasting than saccadic suppression the results would seem to point to a process of post-saccadic visual enhancement, as has been observed in higher brain areas (cf. Ibbotson et al. 2008, 2008). This issue is mentioned but could, in my opinion, be more central to the message of the paper.

**Responses to Reviewer Comments on:
“Saccadic suppression by way of retinal-circuit image processing”
by Idrees, Baumann, Franke, Münch*, & Hafed*
September 24, 2019**

We thank all three reviewers for their insightful comments, which we believe have significantly improved our manuscript. We have now fully addressed all of the reviewer comments, and we have made **major edits** to the manuscript, including the addition of extensive new perceptual and retinal electrophysiology experiments and analyses. For example, all of Figs. 2, 4, 6, as well as Supplementary Figs. 4, 7 now include results from new experiments that we have added for this revision. This was in line with the editorial request, and it was also very helpful in supporting our conclusions and interpretations from the original experiments.

We have also significantly expanded the text length (while still remaining within the editorial guidelines) in order to clarify the significance and novelty of our results.

In what follows, we provide specific responses to the reviewer comments (colored in blue text), and we also highlight the line numbers for the actual edits made in the revised manuscript. The actual manuscript also has relevant edits colored in blue text, in order to assist the editors and reviewers in assessing our revisions.

We sincerely hope that the manuscript is now ready for publication, and we thank the reviewers and editors one more time for the strong encouragement and support to improve the quality of the manuscript.

Reviewer #1 (Remarks to the Author):

REVIEWER COMMENT: In this manuscript, Idrees and colleagues report a series of experiments addressing a contentious question - the mechanisms of saccadic suppression. In human subjects, they find that: (1) detection of luminance increments on low-pass filtered noise textures is impaired around the time of saccades, particularly when a broad blurring filter is used ('coarse'); (2) impairment also occurs when around the time of saccade-like image translations, although the effects are more prolonged; and (3) losses in visibility of grating patches with both real and simulated saccades are spatial frequency selective when surrounded by a coarse (i.e. highly blurred) texture, but spatial frequency invariant when surrounded by a fine (less blurred) texture. In ex vivo retina electrophysiological recordings, they find suppression of responses to luminance increments around the time of simulated saccades, again particularly when presented on highly blurred, 'coarse' textures. The authors conclude that image processing in the retina can account for a large component of saccadic suppression.

Overall, I have mixed feelings regarding this submission. The behavioral experiments comparing real and simulated saccades reveal some potentially interesting results, but I have concerns regarding the methods used to quantify suppression. The electrophysiological result is intriguing and potentially very important. However, it is not very well connected to the third set of behavioral experiments. I've elaborated on these points below.

Thank you very much for this encouragement concerning our manuscript. We appreciate the very careful and useful comments. We have now made a deliberate and faithful effort to respond to all of these comments in the revised manuscript, as detailed below. In particular, we have added new psychophysical experiments complementing our original evidence, and we have also added additional retina electrophysiology experiments improving the links that we would like to highlight to our readership between retinal image processing and perceptual saccadic suppression.

Major:

1. *I have reservations about the use of percent correct detection of a fixed stimulus (luminance increment or grating contrast) to quantify visual sensitivity. For example, the results shown in Figure 1D are interpreted as evidence that saccadic suppression with coarse textures starts earlier and recovers later than with fine textures. However, the floor/ ceiling limits at chance/perfect performance make it very difficult to distinguish a change in the time course of suppression from a change in magnitude.*

We can understand your potential concern about floor and ceiling effects in this experiment, and we have taken significant measures in the revised manuscript to address this point.

Specifically, we have added additional experiments for the revised manuscript, collecting full psychometric curves, exactly as you suggested. In particular, we repeated the experiments of Fig. 1 (and also the simulated saccade version of these experiments; Fig. 5, formerly Fig. 3). However, this time, we used different flash contrasts from trial to trial. Since this was a very data-intensive exercise, we probed only four time points after saccade detection (or four time points relative to simulated saccade onset in the simulated saccade variant of the paradigm). These four time points were chosen strategically to demonstrate all the effects that we were interested in (e.g. perceptual suppression time course and its dependence on background texture statistics). Indeed, the continuity of the suppression profiles that we observed in these new experiments (Figs. 2, 6 showing time courses of elevations of perceptual thresholds consistent with the suppression profiles in Figs. 1, 5), as well as their self-consistency with the results of Figs. 1, 5 make us convinced that our choice of time points was reasonable. As can be seen from the new Figs. 2, 6 and new Supplementary Figs. 4, 7 of the revised manuscript, we not only found results consistent with our original conclusions, but, with the more sensitive approach of full psychometric curves, we could also clearly see that perceptual saccadic suppression was much stronger with coarse background textures than with fine textures at peak suppression (also similar to the retina electrophysiology predictions). This means that you have been correct in your critique: ceiling and floor effects in our original data were hiding some of the effects, and the consequence of visual context is **even more** pronounced than what we were able to see from the original data. Please see lines 147-173, 325-338, Figs. 2, 6, and Supplementary Figs. 4, 7 in the revised manuscript.

Therefore, with the combination of our original (e.g. Figs. 1, 5 in the revised manuscript) and new (e.g. Figs. 2, 6 in the revised manuscript) experiments, we are confident that there are differences in the strength and duration of perceptual saccadic suppression with different background textures. Thank you again for encouraging us to collect these new data.

Finally, we would like to point out that in our original experiments (Figs. 1, 5), we deliberately chose very high contrast flashes to begin with. Specifically, Supplementary Fig. 3a, b of the revised manuscript (formerly Fig. S3A) confirmed that our flashes had a contrast level that was way above the limit of ceiling performance in the baseline condition (i.e. without saccades or simulated saccades). More importantly, flash visibility was identical whether with coarse or fine textures. This meant that in the original experiments (Figs. 1, 5), comparisons of the two background textures were made with equal “baseline” flash visibility regardless of texture type. Therefore, differences peri-saccadically were not due to one condition being closer to floor or ceiling than the other. These interpretations are supported by our other experiments (e.g. the *ex vivo* retinal ganglion cell recordings, and the final set of additional perceptual experiments described above).

2. *These problems are compounded when the magnitude of suppression is expressed as the ratio of percent correct performance relative to baseline (no saccade) some of which are at ceiling or floor (Figure 4). As a result, the spatial frequency tuning profiles reported may not be robust. To exemplify the problem, consider the simple case where suppression shifts an underlying psychometric function rightward along the contrast axis. Calculation of a suppression ratio in this manner will return different values at different contrast levels. While I appreciate that it requires a significant investment of time, the only real solution here is to manipulate luminance/contrast and measure thresholds.*

As stated above, we have now added experiments explicitly measuring thresholds to the revised manuscript (please see lines 147-173, 325-338, Figs. 2, 6 and Supplementary Figs. 4, 7 in the revised manuscript).

Concerning the particular example of the former Fig. 4, now Fig. 7, please note that we did not only report ratios in our original manuscript. In fact, we actually also showed the full underlying spatial frequency curves (e.g. the left panels of the former Fig. 4B, D, E, F; now the left panels of Fig. 7 in the revised manuscript). This is also true for the former Figs. S7 and S8 (now Supplementary Figs. 8-10 in the revised manuscript). In all of these cases, we did show the original data without the ratios. Our choice to use the ratios as additional data plots was made to assist in interpreting the raw curves (i.e. better readability), but the raw curves were always shown. In fact, we avoided plotting the ratios for the highest spatial frequency exactly to avoid issues of robustness due to the low visibility of such a high spatial frequency in baseline. We have now clarified these points in the revised manuscript (please see lines 381-382, 1179-1180, 1187, 1572-1573).

3. *It should be acknowledged that this is not the first study to compare the effects of real and simulated saccades and that some of the findings have been previously demonstrated. In particular, Diamond et al (2000) also reported suppression of targets presented on a patterned background with simulated saccades, that was more prolonged than with actual saccades.*

We apologize for this oversight. In the revised manuscript, we now acknowledge this explicitly. Please see lines 305-308 of the revised manuscript.

4. *The demonstration that retinal ganglion cell responses to a target flash are suppressed around the time of simulated saccades might be the most interesting finding reported in the manuscript. Given the title, I was expecting the authors to go on and provide some insight into how processing in the retina gives rise to this suppression. However, this isn't really developed in the current manuscript. It is also unclear whether any of the results obtained with grating stimuli presented on gray backgrounds could be accommodated by retinal processing. The end result is a manuscript that, despite its substance, feels somewhat disjointed and incomplete.*

We fully understand your point. Given the multi-disciplinary audience that our manuscript caters to, we were initially treading very carefully in our original manuscript, in order to maximize readability for all audiences. The manuscript was also often quite brief in places where important explanation was necessary. In the revised manuscript, we have now followed your advice exactly, in order to address this comment, and we have done so by taking the following measures:

- We have significantly expanded our manuscript's text (while remaining within the editorial limit), allowing us much needed space to fully develop our narrative. In particular, our Introduction is now 537 words, our Discussion is now 1110 words (both much longer than our original Introduction and Discussion).
- As part of the above expansion, we have added a full Results section further developing the retinal mechanisms that we believe underlie perceptual saccadic suppression (please see the Results section starting in line 100 of the revised manuscript).
- We have added substantial new recordings from 376 ganglion cells showing that visual processing, as a result of stimulus-stimulus interactions, causes retinal saccadic suppression (lines 234-284 and Fig. 4). These data show that retinal saccadic suppression arises simply as a result of transient changes across the ganglion cells: whether these transient changes occur due to texture displacements, texture jumps, or simply an intensity change (Fig. 4). Such transient changes, of course, naturally occur during saccades.
- We have clarified why our inclusion of experiments with gray backgrounds was absolutely necessary (in our overall narrative) to demonstrate how wide an array of perceptual phenomena may be visual in origin (for example, please see lines 490-499).
- Finally, we have modified our title to: ***Perceptual saccadic suppression starts in the retina***. We believe that this title is much more descriptive of our results and their novelty.

Minor:

1. *Line 65, note that given a single frame presentation, the actual duration of the localised probe flash would be significantly less than 12ms.*

Yes, we agree. This is a constraint of display technology. We have now clarified this point in the revised manuscript (please see lines 844-851). Our flash duration is also very similar to earlier studies of the kind using similar display technology. Moreover, in these phenomena, it is known that the briefer the probe, the better the effect.

2. *Line 77. The highlighted time intervals in Figure 1 were difficult to see in my version.*

We agree that the gray lines may have been too faint in our original figures. In the revised manuscript, we have rectified this issue (also in all other figures).

3. *Line 168-169. Some referencing inconsistencies here.*

Thank you for pointing this out. We have now proofread all citations.

Reviewer #2 (Remarks to the Author):

Hafed, Idrees and colleagues have measured in isolated retinae of mice and pigs the effect of image motion on responsiveness of retinal ganglion cells. They find properties of “saccadic suppression” similar to those measured in a parallel, behavioral study in human participants. This “saccadic suppression” induced by image motion outlasted suppression induced by saccades. The authors argue to have identified a far-reaching contribution of visual processing mechanisms to saccadic suppression at the level of the retina, which would be in place without motor-based suppression commands, i.e. efference copy or corollary discharge.

This is an exciting study including heroic and state-of-the-art experiments on isolated retinae. Data analysis is fully adequate, results are clear. The fact that neural correlates of effects, which have been described behaviorally here or before, can potentially be found already at the level of the retina, is really exciting. Yet, I do have major concerns about the experimental approach and the interpretation of data, and the conclusions drawn from these results. I detail my concerns below.

We thank you very much for this supportive assessment. We have followed all of your advice, and that of the other two reviewers, in our revised manuscript. We sincerely believe that our manuscript is now significantly improved.

Major:

1. *The experimental apparatus: if I understood it correctly, the stimulus projector was running at 60 Hz (page 32). To my best understanding, this is not adequate. I have to say that I am not an expert of mouse or pig retina, but is it clear that 60 Hz is way above flicker fusion frequencies of retinal cells in these species? If not, I would have a major concern about intra-retinal signal processing during stimulation. In addition: I guess that the presentation of a textured pattern moving at a hundred degrees per second is not easy if not impossible at that low frame rate. The authors must comment on that.*

Thank you for this comment.

In the revised manuscript, we have added text to clarify issues related to display frame rates (please see lines 600-604, 716-735). Briefly, 60 Hz is above the flicker fusion frequency of mouse retina, and it is a standard rate used in many retina electrophysiology laboratories. There is much less information on pig retina in the literature, unfortunately, but our own work suggests that 60 Hz is also above the flicker fusion frequency for this species as well.

More critically, in our manuscript, an important point that we tried to make was that the suppression in the retina does not critically depend on exact saccade velocity profiles. For example, in Fig. 4b, we showed that a simple instantaneous image “jump” from the pre-saccade location to the post-saccade location had the same modulatory properties as the 6-frame simulated movement that we used in the standard experimental paradigm of Fig. 3. In fact, we have now added even more experiments in which we have introduced simple contrast changes and replicated the suppression of retinal ganglion cell responses that we observed with texture displacements (Fig. 4). Therefore, we now have a much more explicit mechanistic explanation in the revised text (e.g. Results section starting at line 234 and Discussion) of how we think that

the retinal contribution to perceptual saccadic suppression arises. Such a contribution does not critically depend on our displayed frame rates in the experiments.

2. *“Saccadic suppression” in retina, Figure 1 : The authors have determined the response for a “probe flash” after presentation of a “saccade like texture displacement”. My first question is: was the texture visible at all times, and only moved at $t=0$ ms. Or was the texture switched on and started moving simultaneously at $t=0$ ms, and switched off after movement. This is critical for the following. Importantly, this texture displacement (or texture onset plus movement) induces retinal activity, as shown in Figure 2B. Hence, the probe-flash can be considered a second stimulus. Previous work (e.g. Mayo and Sommer, Neuronal Adaptation Caused by Sequential Visual Stimulation in the Frontal Eye Field, J.Neurophysiol., 2008) has shown, that at some stages of visual processing, the response to this second stimulus can be clearly reduced if it is presented shortly after the first stimulus. Actually, Mayo’s and Sommer’s Figure 4 is very similar to Figure 2E and 2F of this manuscript. Yet, Mayo and Sommer consider response attenuation for early second stimuli to be an effect of adaptation. Hence, from my point of view, a critical control experiment is missing here. In this control experiment two probe-flashes should be presented, with Inter-Stimulus-Intervals (ISIs) as used in the study for texture and probe. Only, if in such case (i.e. two probe-flashes with different ISIs) the second response would be identical for all ISIs, one can conclude that the movement of the texture caused something like saccade-like suppression. If, however, the response amplitude for the second probe-flash would be very much reduced, then the effect as shown in Figure 2 should be considered an inhibition due to stimulus-repetition rather than “saccadic suppression”.*

Thank you very much for bringing up the very interesting Mayo and Sommer results, which we definitely agree should have been mentioned in our original manuscript. We have now rectified this issue in the revised manuscript (e.g. please see lines 434-446).

Concerning your specific question here, the background was always visible (to both humans and retina) well before and well after the texture displacement at time 0. Therefore, there was no visual transient associated with the background texture onset itself. For the human experiments, the subjects were already steadily fixated with the texture already present, and it was only the displacement that happened as the “visual transient”. The flash was the so-called “second stimulus”. Similarly, in the retina, the texture was always present. The response to the texture displacement that we showed in the original Fig. 2B (now Fig. 3b) was a response to the “displacement” of the texture (i.e. the simulated saccade) and not to an “onset” of the texture. Therefore, the effects that we observed were associated with an interaction between the image translation (the texture displacement) and the flash. We have clarified these points in the revised manuscript (e.g. please see lines 234-284).

We also fully agree with you that stimulus repetition effects are indeed at play here. In fact, this is exactly what we show and claim, and we have even added additional retina electrophysiology experiments (following your and Reviewer 1’s advice) exactly supporting this idea (Fig. 4). Simply put, our objective is to show that perceptual saccadic suppression as it is currently studied in the laboratory setting is fundamentally (at least from a visual perspective) **exactly a stimulus repetition paradigm**: there is the visual transient associated with the global image

motion caused by the saccade (first stimulus) and the visual transient associated with the flash (second stimulus; which could still be presented pre-saccadically). While the first stimulus, caused by the saccade, has some stereotypical properties (namely, a global and fast movement), we find that this is not critical for the suppression to occur (Fig. 4). This is also consistent with our perceptual experiments using “simulated” saccades (Figs. 5, 6).

Thus, fundamentally, our message to the community is that in studies of perceptual saccadic suppression (and even some neurophysiological studies like Berman et al. J. Neurophysiol. 2017 or Chen and Hafed, J. Neurophysiol., 2017), perception (or visual neural activity) is probed using two “stimulus” events: one is the global image motion caused by the saccade itself, and the other is the brief probe flash itself. With this insight, effects like those of Mayo and Sommer should be expected to occur also in the context of saccades. What really surprised us is that this stimulus repetition effect already starts at the very first stage of visual processing (retina), jumpstarting the whole perceptual saccadic suppression process. We find this to be truly amazing! This can also be shown by the suppression of a texture displacement response in the retina by a prior flash response if the flash is presented just before the texture displacement (Fig. 4c). We have now made a stronger effort to clarify these points in the revised Discussion, also relating our work to the rapid neural adaptation effects mentioned by Mayo and Sommer. Please see, for example, lines 434-446, 461-472.

3. *Figure 3: It is already known that the presentation of a textured pattern at high speed induces effects similar to saccadic suppression. This has been shown, e.g. by Diamond, Morrone and colleagues (J.Neurosci., 2000), in their Figure 5. Nevertheless, I agree with the authors that it is exciting to have identified a neural correlate of this behavior at the level of the retina.*

We thank you again for this encouraging comment. Yes, we are aware of these results. As stated to Reviewer 1 above, we apologize for the oversight. In the revised manuscript, we have made our text less brief overall, allowing us to better cite this important result.

4. *Figure 3, continued: also the finding that a saccade-like texture displacement induces effects similar to saccadic suppression, which last longer than saccadic suppression induced by real eye movements, is already known. As an example, see Figure 5 of Diamond et al., J.Neurosci., 2000. These authors mentioned “More remarkable is that the recovery of sensitivity is slower for mirror motion than for saccades” (citation from their paper). So, the conclusion drawn by Diamond and colleagues matches the idea of the authors of the current study.*

We fully agree that the results of Diamond et al do not contradict our work, and we have now clarified this point much more clearly in our revised manuscript (e.g. lines 305-308). Unfortunately, the visual contributions that are so clearly present in their own data have become somewhat masked in the literature in favor of the “extra-retinal” hypothesis. It is our sincere hope that our results will help reinvigorate the idea that early visual processing, already in the retina itself, is rich and robust enough to support perception even from the very get-go. Of course, this does not in any way negate the need for motor-related signals; rather, we very much hope that our insights from the current study will spur novel takes on the role of signals

like corollary discharge. In the revised manuscript, we discuss these points more extensively (e.g. please see lines 513-527).

Minor:

1. *Thiele et al., Science, 2002, should also be cited when referring to previous studies on saccadic suppression (page 1 and 2)*

We agree. Our revised manuscript is now less brief, allowing us to better treat the literature and avoid such oversight. Please see lines 51-53 of the revised manuscript.

2. *Even more important would be to mention already early on and explicitly “masking” as alternative basis of saccadic suppression compared to mechanisms like efference copy or corollary discharge, i.e. in the first paragraph of page 2. I suggest to cite work from Wurtz and colleagues, e.g. Campbell & Wurtz, 1978, but also the excellent review: Wurtz, Vis Res, 2008.*

We have now followed this advice exactly, and we have added citations to Campbell & Wurtz, 1978 and Wurtz, 2008 in our revised Introduction. Please see lines 67-69 of the revised manuscript. We have also discussed visual masking at length in our paper (e.g. lines 474-488).

3. *Lines 112, 113: given my concerns above, sentences like “Rather, suppression was a result of specific spatio-temporal retinal-circuit image processing initiated by the texture displacements” might not be true.*

As clarified above, and in the revised manuscript (e.g. lines 234-284), we think that we agree with you about stimulus repetition effects. This is exactly our point. We have also added additional retina experiments in Fig. 4 supporting this idea: from the perspective of visual processing, what is called saccadic suppression in perceptual experiments is actually fundamentally a two-stimulus paradigm for the visual system (one stimulus is retinal image motion caused by eyeball rotation and another stimulus is the brief flash used to probe perception). Therefore, suppression is in fact a result of specific spatio-temporal retinal circuit image processing initiated by texture displacements. We have, however, revised our wordings in the revised manuscript (e.g. please see lines 277-280).

4. *Line 126: like myself, many readers might not be aware of the native oculomotor behavior of mice and pigs. Authors should briefly explain how often they make saccades, at what max speed, etc.*

Thank you. We have now included more details about the oculomotor behavior of mice and pigs. Please see lines 720-735.

5. *When referring to Castet and colleagues concerning intrasaccadic perception of visual motion, I suggest to refer also to an earlier work on the same issue: Ilg and Hoffmann, Vis Res, 1993.*

Done. Please see lines 67-69 of the revised manuscript.

6. *The authors should check their reference list. As an example, a Castet et al. paper is listed twice.*

We apologize for this issue. We have now double-checked all citations.

Reviewer #3 (Remarks to the Author):

The present study makes two contributions to the discussion. First, it compares the influence of texture patterns on saccadic suppression and masking in humans. Second, it records retinal responses to saccade-like motion of texture patterns and flashes in mouse and pig retina. In comparing the findings from these experiments it shows some strong similarities between saccadic suppression and visual masking but also some clear differences.

We thank you for all the valuable feedback and for this very careful reading of our manuscript! We have incorporated your comments and, below, provide responses in brief.

1. *In the human experiments, participants made saccades across a background of fine or coarse texture while a brief luminance pedestal was presented around the onset of the saccade. The results (Fig. 1) show that suppression begins earlier and lasts longer for the coarse than for the fine texture background. Peak suppression, on the other hand, was not different. Fig. 3 shows the results of a similar experiment with simulated saccades, i.e., a saccade-like displacement of the texture pattern. In this case, the onset of suppression (at least for the coarse pattern) is even earlier and suppression lasts even longer than for the real saccades in Fig. 1. This is clear indication of strong masking in these conditions, and of a difference in masking between the coarse and the fine textures. Unfortunately, there is no data presented for the interesting time shortly before and at motion onset, which is the time of peak suppression. Thus, it is unclear whether real and simulated saccades differ at that time. Yet, the longer time course of masking suggests that suppression is somewhat different from masking.*

We understand your point, and we have taken measures to address it. Specifically, in the revised manuscript, we have now added additional experiments sampling more time points in the simulated saccade perceptual experiments, including time points very near to the onset of the texture displacement (please see the new Figs. 5, 6 and lines 152-160, 927-931). These additional experiments and time points demonstrate an expected temporal continuity of the suppression profile (i.e. as a function of time).

Similarly, for the retina electrophysiology experiments, we did indeed sample near the onset of texture displacement, as shown in the revised manuscript. For example, across all experiment variants in retina electrophysiology sessions, we sampled as close as 1 display frame (~17 ms) from the transient onset “simulating” saccades (e.g. Fig. 4).

As we state in the manuscript, perceptually, real and simulated saccades are fundamentally similar in terms of suppression because we do get the same pre- and post-saccadic suppression whether a real or simulated saccade occurred (please see Figs. 5, 6). Neuronally, we believe that pre-saccadic perceptual effects (for example, in our simulated saccade paradigms with the human subjects) may be mediated by visual mechanisms downstream of the retina (please see lines 320-323, 474-488). Our overall message to the community is that visual-only mechanisms (sometimes called masking) account for a surprisingly large component of the properties of perceptual saccadic suppression, and that such visual-only mechanisms start in the retina. Of course, we do not claim that a motor-derived signal with real saccades

does not play any role. To the contrary, we argue that real saccades (i.e. with a motor-derived signal) have a much shorter period of “suppression”.

2. *Fig. 4 shows a comparison of real saccades and saccade-like image motion for masking by a surrounding pattern. Unlike in Experiment 1, the suppression during saccades is much stronger than the suppression produced by texture displacement (cf Fig. 4B vs 4D and 4E vs 4F). Therefore, this experiment would seem to show some similarity between masking and suppression but also that suppression requires more than just masking.*

Please note that in this figure and experiment (now Fig. 7), the time point sampled for the simulated saccade version of the experiment was substantially later than the time point sampled for the real saccade version; that is, due to the data-intensive nature of the experiment, we reduced the sampled time points. Therefore, it is expected that the suppression effect might be somewhat weaker for the simulated saccade. This is also supported by the time course data in Supplementary Fig. 8.

3. *The experiments on mouse and pig retina measure the influence of saccade-like retinal image motion on RGC responses to brief flashes. Similar to the human suppression/masking data coarse texture patterns produce more masking than fine texture patterns. This would suggest that part of the masking properties originate from the retina. However, the time course of the retinal effects is vastly different from that of saccadic suppression in humans. The retinal effects last for several hundred milliseconds while the suppression during real saccades in humans last only for about 50-100ms after saccade onset. Moreover, the retinal effects seem to produce only forward masking, i.e. the reduction of responses to flashes presented after motion onset. Responses to flashes presented before motion onset, i.e. in a backward masking situation, appear only weakly modulated (Fig. 2E). Like in the human data of Fig. 3 there are not data points at or near motion onset so it is difficult to compare to the most interesting time range for saccadic suppression, i.e., slightly before and at saccade onset.*

As stated above, we had flashes very close to the simulated saccade in our retinal experiments, and also in our new additional experiments summarized in the new Fig. 4.

Also, yes, the retinal suppression indeed lasted for quite a long time after texture displacement. This is exactly why we wanted to emphasize that perceptual suppression in humans without real saccades, but with only texture displacements, also outlasts “real” saccadic suppression (Figs. 5, 6). So, there is indeed strong agreement between our retinal and perceptual results.

We are reluctant to delve deeper into quantitative comparison of “time to recovery” between perception and electrophysiology because such quantitative comparison would be too assumption-laden. Specifically, in the retinal data, we have analyzed ganglion cell response

strength (peak of spiking response), and found it being suppressed for a long time. But, even a retinal response that is reduced by 20 or 30% may ultimately lead to a clear supra-threshold perception of the underlying stimulus. Therefore, the time course of perceptual recovery may be faster than the time course of neural recovery. We simply do not know the relationship between retinal response strength and perceptual threshold, so that the overly long time course in retinal suppression might be a property of the exquisite analytical detail that is possible for electrophysiological data. What is encouraging, however, and highly consistent with the retinal data, is the prolonged perceptual suppression with simulated (rather than real) saccades, as stated above.

Finally, for simulated saccades and flashes before displacement onset, we mentioned how we found perceptual suppression in humans but not convincing suppression in the retina. This would be classified as an instance of “backwards masking” in perception. It is still visual in origin, but we suggest that it likely happens in visual areas downstream of the retina (please see lines 320-323, 474-488).

In summary, the paper makes some interesting contributions on visual masking, its origin in the retina, and its contribution to the impairment of visual detection during a saccade. Since the data from before and near saccade/motion onset is either missing or inconsistent the conclusion that explicit motor-based suppression is not needed is too strong. Since in some conditions visual masking is longer lasting than saccadic suppression the results would seem to point to a process of post-saccadic visual enhancement, as has been observed in higher brain areas (cf. Ibbotson et al. 2008, 2008). This issue is mentioned but could, in my opinion, be more central to the message of the paper.

We thank you for all of your comments. We have now followed your advice concerning motor-based versus visual mechanisms, and also the interesting post-saccadic enhancement results of Ibbotson and others. Please see lines 513-527 of the revised manuscript. Briefly, we made sure to clarify that we do not at all fully negate the use of motor-based mechanisms. We think that they contribute, but in different ways than is generally assumed in the literature. We also definitely agree about the Ibbotson studies.

Concerning times near motion onset, as stated above, we did indeed show results before and near motion onset in the simulated saccade paradigm in both perception and retina electrophysiology (e.g. Figs. 4-6 and Supplementary Figs. 5, 7), and our results between retinal recordings and perception were consistent overall. The interesting observation here was that perception showed very clear suppression before motion onset but retina less so (if one simply measures the flash response that happened before the texture displacement). It is really interesting to investigate in future studies the mechanisms underlying such “backwards masking” effect in perception. As far as we know, this is a robust perceptual phenomenon (backwards masking) but with no convincing neural substrates.

Reviewers' comments:

Reviewer #1 (Remarks to the Author):

The authors have done a good job of responding to my queries and improving their manuscript. In particular, their demonstration that key outcomes of human experiments 1 & 2 are replicated when thresholds are derived from full psychometric function makes for a more convincing case.

As noted in the authors' response, this additional data revealed that differences in suppression profile between visual context conditions were actually bigger than initial measurements of detection performance for a fixed contrast stimulus suggested. With this in mind, I would strongly recommend they carry this approach forward to the final human experiment as well. That manipulations of a surrounding texture alter the spatial frequency tuning of saccadic suppression is an intriguing finding that I suspect will pique the interest of many working in the field. However, forming strong conclusions about tuning requires confidence in the measurements of suppression at each spatial frequency and, as outlined in my original review, ratios of percent correct relative to varying baselines can be problematic. While I appreciate that collecting full psychometric functions is non-trivial, I suspect it would be a worthwhile investment of time.

Some details regarding the method used to fit psychometric functions and derive thresholds would also help. Some of the panels in Supplementary Figure 7 indicate that a lapse parameter was used - was this free to vary in each fit? Also, while the main text suggests that thresholds were defined at a fixed performance level (62.5%), they are shown to correspond to the point of inflection of the function in Sup. Fig. 7.

Line 950. 'This allowed filling [fitting] the psychometric curves'

Reviewer #2 (Remarks to the Author):

I am impressed by the work the authors have invested in order to revise the manuscript. As recommended by the reviewers and the editor, the authors have conducted additional behavioral and neurophysiological experiments. From my point of view, they have answered to most of the concerns convincingly. There is one point, however, which I am not happy with: double pulse stimulation and saccadic suppression, from my point of view, are two different things. The authors show that the reduction in responsiveness for the probe stimulus does not require a previously shown pattern to move. Instead, it is the brief interval between any first and any second stimulus which induces this response reduction. And the time course of recovery is very similar to results on double pulse stimulation shown in the literature before.

I agree that many of the results are interesting and especially the finding that many of the effects which can be observed at the cortical level are similarly present already in the retina, is exciting. Yet, from my point of view, this has not necessarily something to do with saccadic suppression. Therefore, I do not agree with the statement of the authors from their rebuttal letter: "Simply put, our objective is to show that perceptual saccadic suppression as it is currently studied in the laboratory setting is fundamentally (at least from a visual perspective) exactly a stimulus repetition paradigm".

Vision is continuous, as is natural visual stimulation. Stimulation in the lab is discontinuous. This is a critical difference. And as stated in my review of the original version of the manuscript: "If, however, the response amplitude for the second probe-flash would be very much reduced, then the effect as shown in Figure 2 should be considered an inhibition due to stimulus-repetition rather than "saccadic suppression".

Reviewer #3 (Remarks to the Author):

In this revision, the authors have done a commendable job adding further experiments and clarifying many issues. The paper contains a wealth of very valuable data showing many similarities between real and simulated saccades in humans, and between simulated saccades in human perception and retinal physiology. However, I still think that the authors stretch their conclusions at some points. Given the confusions in the literature, which the authors themselves rightfully bemoan, I think one should be as clear and neutral as possible in the presentation of the implications of this study. I list my remaining concerns below.

Major:

1. It starts in the abstract. The authors claim that visual-only mechanisms, activated by saccade-induced image shifts, can account for ALL perceptual properties of saccadic suppression that they have investigated. This is not true. The time-course is different (real saccades show shorter suppression than simulated saccades), the retinal data lack pre-saccadic suppression, and there is a further issue detailed below in point 2.

2. This issue concerns the data from Fig. 4. It shows modulation indices for texture jumps and contrast steps. Both look considerably different from what would be expected for saccadic suppression. Near time zero, which would correspond to saccade onset, the modulation index is much smaller than at later time points, for example around 50ms. In saccadic suppression, in contrast, modulation is largest at saccade onset.

3. I think this has to do with the mechanism the authors suggest for the retinal effects. Basically, as far as I understand, it consists of an interaction between two successive visual stimuli in which the retinal response to the second stimulus is weakened by the first. What happens if both come at the same time (0ms or saccade onset)? Do they perhaps combine? It is hard to see how this relates to saccadic suppression. There is also a problem when the order of presentation is reversed (lines 253 - 256 and Fig. 4c). In this case the response to the second stimulus, the displacement, is suppressed. On the face of it this should predict that one now perceives the flash correctly but the texture displacement only weakly. Clearly, this is inconsistent with saccadic suppression in which one does not perceive the flash presented before the saccade, and actually never perceives the texture displacement.

Let me be clear here: I do not doubt that this mechanism exist and I commend the authors for describing it in so much detail in the wealth of data they collected. I am also struck by its dependencies on spatial frequency. I just want to point out that it provides only a partial match with all of saccadic suppression and that it contains some logical inconsistencies.

Minor:

The contrast step experiments suggest that a contrast transient produces a similar weakening of the retinal response to a subsequent flash as the texture displacement. Does this also occur in perception? Can the authors explain a bit more their prediction that a high-contrast visual transient would be equivalent to the displacement of a coarse texture across a given RGC receptive field, while a low-contrast transient would be equivalent to the displacement of the fine texture across the same receptive field. It is not readily clear to me why the motion energy expected with each type of texture displacement is different.

Lines 305 - 308: This is confusing when presented in such a short way.

Lines 401 - 404: This seems to apply only for stimuli presented after saccade onset, not for stimuli presented before or at saccade onset.

**Responses to Reviewer Comments on:
“Perceptual saccadic suppression starts in the retina”
by Idrees, Baumann, Franke, Münch*, & Hafed*
January 28, 2020**

We thank all three reviewers for their new comments. We have now fully addressed all of the remaining points, and we have made corresponding edits to the manuscript, including the addition of extensive new perceptual experiments and analyses. We have also reworded sections of the manuscript to make our results, interpretations, and conclusions clearer. This was in line with the editorial request, and has improved our manuscript significantly.

In what follows, we provide specific responses to the reviewer comments (in blue text), and we also highlight the line numbers for the actual edits made in the revised manuscript. The actual manuscript also has relevant edits colored in blue text, in order to assist the editors and reviewers in assessing our revisions.

We sincerely hope that the manuscript is now ready for publication, and we thank the reviewers and editors for the strong encouragement and support to improve the quality of the manuscript.

Reviewer #1

REVIEWER COMMENT: The authors have done a good job of responding to my queries and improving their manuscript. In particular, their demonstration that key outcomes of human experiments 1 & 2 are replicated when thresholds are derived from full psychometric function makes for a more convincing case.

As noted in the authors' response, this additional data revealed that differences in suppression profile between visual context conditions were actually bigger than initial measurements of detection performance for a fixed contrast stimulus suggested. With this in mind, I would strongly recommend they carry this approach forward to the final human experiment as well. That manipulations of a surrounding texture alter the spatial frequency tuning of saccadic suppression is an intriguing finding that I suspect will pique the interest of many working in the field. However, forming strong conclusions about tuning requires confidence in the measurements of suppression at each spatial frequency and, as outlined in my original review, ratios of percent correct relative to varying baselines can be problematic. While I appreciate that collecting full psychometric functions is non-trivial, I suspect it would be a worthwhile investment of time.

Thank you very much for the encouraging remarks. In line with your advice, we have now performed additional experiments collecting full psychometric curves to explore the spatial frequency tuning of perceptual saccadic suppression. We did this for real saccades as well as for simulated saccades. Specifically, our original experiments (Fig. 8 in the revised manuscript) showed that baseline visibility was at ceiling performance for the three lowest tested spatial frequencies. Importantly, for these same spatial frequencies, peri-saccadic visibility had different tuning properties in the different surround contexts (our original results; Fig. 8). Therefore, focusing on the lowest 3 spatial frequencies in our new experiments (when collecting full psychometric curves) was sufficient to balance the conflicting needs of:

(1) showing differential suppression as a function of spatial frequency with full psychometric curves; and (2) making the numbers of needed trials in the new experiments practically manageable (since full psychometric curves are data intensive endeavors). Therefore, we used 3 spatial frequencies as the peri-saccadic “flashes”, 2 surround contexts (coarse and fine), 2 eye movement types (real saccades and simulated saccades), 2 strategically-chosen sampling times around the eye movements (to show perceptual recovery long after the movements), and >5 different contrasts for each spatial frequency (to allow fitting full psychometric curves). The results for the real saccades are shown in the new Fig. 9, and the results for the simulated saccades are shown in the new Fig. 10. As can be seen, we validated all of our original findings: relative to baseline visibility (which was similar for all 3 spatial frequencies and surround contexts), there was clear selectivity of saccadic suppression for both real and simulated saccades when the surround context was coarse, and there was no selectivity when the surround context was fine. That is, detection thresholds were different as a function of spatial frequency only in the coarse surround condition (with or without real saccades). These results directly support our original findings (Fig. 8). Please also see lines 449-455.

Some details regarding the method used to fit psychometric functions and derive thresholds would also help. Some of the panels in Supplementary Figure 7 indicate that a lapse parameter was used - was this free to vary in each fit? Also, while the main text suggests that thresholds were defined at a fixed performance level (62.5%), they are shown to correspond to the point of inflection of the function in Sup. Fig. 7.

Yes, the lapse parameter was free to vary in each fit. We have now clarified this in lines 1295-1299 of the revised manuscript.

We also apologize for the lack of clarity on the calculation of perceptual thresholds. We have now remedied this in lines 1302-1306. Briefly, the threshold was defined for each individual subject's psychometric curve while taking into account the lapse rates. That is, the threshold was set at 62.5% of the dynamic range of the individual subject's psychometric curve (as recommended by the Psignifit authors). This is why the horizontal error bars were at different y-axis values in Supp. Figs. 4 and 7. When collecting population summaries of the thresholds (e.g. Figs. 2e, 7e, 9c,d, and 10c, d), we first calculated each subject's individual threshold and then pooled to provide a representative population summary. The pooled psychometric curves (e.g. Fig. 2a-d) were also averages of the individual subject psychometric curves (to visualize the overall consistency of the results), and we removed any explicit fixed performance level in these pooled curves (i.e. the old 62.5% horizontal lines) as the detection threshold to avoid any confusion.

We have now checked all figures with psychometric curves as per the above.

Line 950. 'This allowed filling [fitting] the psychometric curves'

We have now modified the sentence, as suggested. Please see line 1027 of the revised manuscript.

Reviewer #2

I am impressed by the work the authors have invested in order to revise the manuscript. As recommended by the reviewers and the editor, the authors have conducted additional behavioral and neurophysiological experiments. From my point of view, they have answered to most of the concerns convincingly. There is one point, however, which I am not happy with: double pulse stimulation and saccadic suppression, from my point of view, are two different things. The authors show that the reduction in responsiveness for the probe stimulus does not require a previously shown pattern to move. Instead, it is the brief interval between any first and any second stimulus which induces this response reduction. And the time course of recovery is very similar to results on double pulse stimulation shown in the literature before.

I agree that many of the results are interesting and especially the finding that many of the effects which can be observed at the cortical level are similarly present already in the retina, is exciting. Yet, from my point of view, this has not necessarily something to do with saccadic suppression. Therefore, I do not agree with the statement of the authors from their rebuttal letter: "Simply put, our objective is to show that perceptual saccadic suppression as it is currently studied in the laboratory setting is fundamentally (at least from a visual perspective) exactly a stimulus repetition paradigm:".

Vision is continuous, as is natural visual stimulation. Stimulation in the lab is discontinuous. This is a critical difference. And as stated in my review of the original version of the manuscript: "If, however, the response amplitude for the second probe-flash would be very much reduced, then the effect as shown in Figure 2 should be considered an inhibition due to stimulus-repetition rather than "saccadic suppression".

We thank the reviewer very much for the encouraging remarks.

We have taken significant measures in the revised manuscript to clarify our points, and we summarize these measures below (along with brief excerpts of the arguments introduced in the revised manuscript):

- 1) We fully agree that vision is continuous. In fact, our retinal results show that vision is continuous because there are clear and strong responses to saccade-induced image shifts, as opposed to suppression (Fig. 3b). What is very interesting from our perspective is that saccades do actually introduce clear temporal "events" that are characterized by strong, but short-lived, retinal transients (Fig. 3b); it is exactly the interaction of these discrete temporal "events" with flash events that is a stimulus-repetition scenario from a visual perspective. Thus, classic studies of perceptual saccadic suppression (saccadic eyeball rotation coupled with a brief visual probe) are indeed the same, from a visual perspective, as stimulus-repetition paradigms (one of the stimuli just happens to be caused by eyeball rotation). We have further clarified this point in lines 482-501.
- 2) Related to the above, we reiterate that in all of our "simulated" saccade paradigms (both in electrophysiology and psychophysics), the background texture was continuously present all the time to ensure that the only transients that occurred across the retina were due to texture displacements mimicking the transients across the retina during real saccades. Please see lines 812-814 and 954.

- 3) We also fully agree that the expression “saccadic suppression” is sometimes overloaded in the literature. Classically, the phenomenon of reduced perceptual sensitivity for brief peri-saccadic flashes is known as “saccadic suppression”; this is exactly how the expression is used most frequently in the literature, and this is how we defined it in the very first paragraph of our Introduction (lines 57-60). Similarly, the phenomenon of reduced flash perception when presented in temporal proximity to a masking stimulus is termed as “visual masking”, and the phenomenon of reduced flash neural response when presented after another flash is termed as “neural adaptation”. The suppression of flash response (or perception), often noted as an increase in contrast threshold, in all of these three seemingly different paradigms is actually qualitatively similar. This is, in fact, not surprising because the image flow across the retina in all of these paradigms is similar: a first stimulus (e.g. saccade / mask / flash) induces a visual transient, and this is paired with a test flash inducing a second visual transient. Despite this, the resulting suppression is termed differently depending on the paradigm used. Naturally, it is fine to call suppression occurring during saccades as saccadic suppression and suppression occurring by masks as visual masking, as a way of clarifying the context of the phenomenon. However, what we are merely demonstrating in our study is that visual mechanisms triggered by stimulus-stimulus interactions underlie all of these phenomena. Moreover, such visual mechanisms are already triggered at the level of the retina since visual transients across the retina, caused by image shifts, naturally occur during eye movements. Please see lines 482-501. Of course, we emphasize again that this does not deny the existence of extra-retinal mechanisms associated with saccades. In fact, we argue that motor-related signals might play a critical role in our very same paradigms (e.g. shortening of suppression).
- 4) In line with the above, we have added new experiments (Fig. 5) demonstrating remarkably similar perceptual suppression just in response to background luminance changes (as in our retinal experiments of Fig. 4d) and without any saccades or image displacements. The perceptual phenomena across all paradigms (real saccades, simulated saccades, or just background luminance transients) look almost indistinguishable from each other, and this is exactly our point. Please see Fig. 5 and lines 286-298.
- 5) Finally, we have added text in the discussion (lines 576-586) to clarify the difference between the terms “saccadic omission” and “saccadic suppression” in the literature. Saccadic omission is used to describe the phenomenon of how background image motion (in an otherwise stationary scene) is not perceived across saccades (e.g. if we make a saccade across a stationary image, we do not perceive the intra-saccadic motion caused by the saccade). On the other hand, the term saccadic suppression is used in the literature when brief visual probes are flashed around the time of saccades (the motivation being that brief probes measure the momentary sensitivity of the visual system). It is exactly this phenomenon that we focused on for this study. As we clearly show, these brief visual probes (and the suppressed percepts that they are associated with) are in fact part of stimulus-repetition scenarios from a visual perspective. We emphasize that we used exactly the same saccadic suppression paradigms as in the literature; we did not invent the paradigm, but we merely studied it from the perspective of visual (and retinal) transients. Please see lines 482-501.

Reviewer #3

In this revision, the authors have done a commendable job adding further experiments and clarifying many issues. The paper contains a wealth of very valuable data showing many similarities between real and simulated saccades in humans, and between simulated saccades in human perception and retinal physiology. However, I still think that the authors stretch their conclusions at some points. Given the confusions in the literature, which the authors themselves rightfully bemoan, I think one should be as clear and neutral as possible in the presentation of the implications of this study. I list my remaining concerns below.

Thank you once again for the encouraging remarks and the recognition. We have now followed your advice to make our text clearer. We have also added new experiments to provide further support.

Major:

1. It starts in the abstract. The authors claim that visual-only mechanisms, activated by saccade-induced image shifts, can account for ALL perceptual properties of saccadic suppression that they have investigated. This is not true. The time-course is different (real saccades show shorter suppression than simulated saccades), the retinal data lack pre-saccadic suppression, and there is a further issue detailed below in point 2.

We have made this sentence clearer, and we have also introduced other clarification points throughout the manuscript (e.g. Discussion). Please see lines 42, 331-348, 365-369, 521-526, and 568-578 of the revised manuscript.

Briefly, we do show that visual mechanisms (both retinal and post-retinal) underlie all of the perceptual properties of saccadic suppression that we investigated. By “perceptual properties”, we mean properties related to image processing (e.g. dependence on texture statistics). For example, the suppression that we observed in our perceptual experiments depended on whether the background texture was coarse or fine. Critically, the dependence was identical (coarse causing stronger and longer lasting suppression than fine even with simulated saccades) whether there was a real or a simulated saccade (and also identical to how the retinal output was suppressed as a function of texture statistics). This could only be the case if visual mechanisms played an important role in saccadic suppression.

Furthermore, the selectivity of perceptual suppression for gratings of low spatial frequency (Burr et al., 1994) also occurred in the absence of any saccadic eye movements; and, we could easily modulate the selectivity with or without real saccades by simply changing the visual context. In short, we investigated a wide array of perceptual saccadic suppression properties, and indeed, our results show that visual mechanisms can account for all of these properties.

Concerning pre-saccadic suppression, please note that we did indeed observe “pre-saccadic” suppression in the psychophysical experiments with simulated saccades (please also see the new Fig. 5 experiments for further evidence for “acausal” suppression in the absence of real saccades). This means that visual mechanisms still contributed to explaining pre-saccadic suppression with real saccades, even if this particular aspect of the

phenomenon might arise from neural sites subsequent of the retina. In other words, our original abstract sentence mentioned “visual mechanisms” and did not really claim that retinal image processing accounted for pre-saccadic suppression. We still believe that visual processing accounts for pre-saccadic suppression, but, as we state in our manuscript (e.g. lines 331-348 and 516-530), this might arise through other visual-only mechanisms beyond the retina.

Concerning time courses, yes, the presence of saccadic eye movements in our experiments revealed that eye movement related signals shortened both pre- and post-saccadic suppression caused by visual mechanisms. Therefore, if anything, our results demonstrate an opposite role of eye movement signals in saccadic suppression than what is currently believed in the literature. However, in our view, the time course of the suppression is distinct from what we consider to be the “perceptual properties” of suppression.

Please also see our response below for the remaining point that you mentioned.

2. This issue concerns the data from Fig. 4. It shows modulation indices for texture jumps and contrast steps. Both look considerably different from what would be expected for saccadic suppression. Near time zero, which would correspond to saccade onset, the modulation index is much smaller than at later time points, for example around 50ms. In saccadic suppression, in contrast, modulation is largest at saccade onset.

We agree that the time for peak suppression in the retinal data is slightly later than the peak suppression time seen perceptually. However, this is not grounds to reject the notion that visual mechanisms, starting in the retina, are part of the mechanisms for saccadic suppression. Specifically, and as stated above, at the perceptual level, we see suppression starting already **before** the simulated saccades (please also see the new Fig. 5 for yet another example of such “acausal” effect in perception and lines 331-348). This suppression, probably occurring through backwards visual masking, could interact with the post-saccadic suppression coming from the retina to give rise to the observed perceptual time course (which would necessarily deviate slightly from the retinal suppression time course). More studies will have to be undertaken to understand in detail backwards masking and its interaction with the visual input from the retina. However, once again, this point does not change the fact that suppression is (at least partially) mediated by visual mechanisms because, as stated in point 1 above, strong suppression still occurs in the absence of eye movements and depends on the visual input during both real or simulated saccades. Furthermore, this dependence on visual input starts already in the retina. Please see lines 467-480.

3. I think this has to do with the mechanism the authors suggest for the retinal effects. Basically, as far as I understand, it consists of an interaction between two successive visual stimuli in which the retinal response to the second stimulus is weakened by the first. What happens if both come at the same time (0ms or saccade onset)? Do they perhaps combine? It is hard to see how this relates to saccadic suppression.

In the case where two stimuli completely overlap, we will not have a way to test the visual sensitivity of the system. If we then introduce a third stimulus with a temporal delay, it will experience the combined effect of the first two stimuli, which will in fact be equivalent to a single stimulus onto the retina. The relationship to saccadic suppression is rooted exactly in the idea of considering a purely visual perspective of oculomotor behavior: a first stimulus causes a visual transient, triggering an image processing cascade in the retina, that influences the response to the second stimulus. This first stimulus can be a mask, a flash, or image shifts such as those that naturally occur during saccades. We do not claim that this visual mechanism is itself saccadic suppression, since the term saccadic suppression describes a perceptual phenomenon. Rather, we show that visual mechanisms, starting already in the retina, contribute towards causing this perceptual phenomenon of saccadic suppression. Please see lines 482-501. Furthermore, we do not dispute that other mechanisms might also be involved.

There is also a problem when the order of presentation is reversed (lines 253 - 256 and Fig. 4c). In this case the response to the second stimulus, the displacement, is suppressed. On the face of it this should predict that one now perceives the flash correctly but the texture displacement only weakly. Clearly, this is inconsistent with saccadic suppression in which one does not perceive the flash presented before the saccade, and actually never perceives the texture displacement.

It is still not entirely clear from the literature whether one can perceive the background during saccades or not. Some studies have reported not seeing the background during saccades and have termed it as saccadic omission (Campbell and Wurtz, 1978; Ibbotson and Cloherty, 2009). However, there are psychophysical studies showing that it is actually quite easy to perceive the background during saccades as long as the image flow across the retina is optimized for the motion sensitive system (Castet and Mason, 2000; Garcia-Perez and Peli, 2001). These observations, together, show that suppression can depend strongly on the properties of the visual input. This is consistent with our findings that suppression strongly depends on the image statistics of the background. We hope that our results will motivate the investigation of related phenomena like saccadic omission and visual masking under conditions of varying stimulus statistics.

Concerning the scenario of order reversal, we have added text in lines 532-542 exploring possible explanations for it. Given that perception necessarily involves an interpretation of the sensory evidence that depends on priors, one possibility is that certain priors associated with global motion (e.g. that it is a strong indicator of the occurrence of saccades) can cause perception to “omit” the pre-saccadic flash (even though it evokes a strong retinal transient) exactly because of its pairing with strong global motion (even if its neural transient in the retina is weakened by the prior flash). Once again, please see lines 532-542 for more details.

Let me be clear here: I do not doubt that this mechanism exist and I commend the authors for describing it in so much detail in the wealth of data they collected. I am also struck by its dependencies on spatial frequency. I just want to point out that it provides only a partial match with all of saccadic suppression and that it contains some logical inconsistencies.

Thank you very much for appreciating our work and for your valuable comments. We have now addressed this point further in the discussion section. Please see lines 516-530 and 568-576 of the revised manuscript.

Briefly, and as stated above, we believe that the inconsistencies pointed out above are not really inconsistencies (please see our earlier responses). However, we do acknowledge that whatever remains (i.e. the exact mechanisms for pre-saccadic suppression) is ultimately due to gaps in our knowledge about the neural mechanisms of backwards visual masking. There is simply no convincing neural mechanism for backwards visual masking in the literature, and we think that this is a critical point for future research. This phenomenon is so robust perceptually, and it also appears in our newest experiments (see Fig. 5). Yet, how it arises from either retinal or post-retinal neural processing is still unknown. We certainly hope that our manuscript will motivate research in this direction, by us and others.

Similarly, it would be very interesting to understand how motor-related signals shorten the duration of perceptual suppression in our paradigms.

At this point, the purpose of our manuscript is to document (with as much convincing experimental evidence as possible) the fact that a wide array of perceptual saccadic suppression phenomena can be explained by visual mechanisms. We sincerely hope that our study stimulates the right questions to further understand how signals from different areas and modalities interact to render stable perception during natural vision.

Minor:

The contrast step experiments suggest that a contrast transient produces a similar weakening of the retinal response to a subsequent flash as the texture displacement. Does this also occur in perception?

Yes. We have now performed new experiments directly addressing this point. Subjects simply fixated, and we created a step in the background luminance. At a random time relative to the contrast step, we presented our usual detection probe as in our original experiments. There was clear perceptual suppression, which clearly depended, in its strength, on the size of the luminance step (i.e. whether it was a high or low contrast step). This is directly consistent with the retinal results. Please see the new Fig. 5 and lines 286-298 of the revised manuscript. Also, please note that we have changed the term referring to these experiments from “contrast step” to “luminance step” to avoid any confusion in understanding our experimental manipulation.

Can the authors explain a bit more their prediction that a high-contrast visual transient would be equivalent to the displacement of a coarse texture across a given RGC receptive field, while a low-contrast transient would be equivalent to the displacement of the fine texture across the same receptive field. It is not readily clear to me why the motion energy expected with each type of texture displacement is different.

We have added more details on how a high contrast step would be equivalent to displacement of a coarse textured background. Please see lines 1250-1260.

Briefly, from the perspective of visual transients across the retina, low contrast steps are equivalent to fine texture displacements over receptive fields, and high contrast steps are equivalent to coarse texture displacements. This is simply because of the spatial relationship between receptive field sizes and texture spatial scales: a fine texture presents both dark and bright blobs within individual receptive fields both before and after the texture displacement (resulting in a low contrast change in luminance over the receptive fields); on the other hand, a coarse texture has dark or bright blobs that are of similar size to the receptive fields (resulting in the potential for a very large contrast change in luminance over the receptive fields after the texture displacement).

Lines 305 - 308: This is confusing when presented in such a short way.

We have now simplified the text to restrict it to just describing empirical observations and therefore avoiding potential confusions. Please see lines 328-329.

Lines 401 - 404: This seems to apply only for stimuli presented after saccade onset, not for stimuli presented before or at saccade onset.

We are not sure that we understand this comment. We did study pre-saccadic suppression in our experiment with Gabor gratings of different spatial frequencies.

REVIEWERS' COMMENTS:

Reviewer #1 (Remarks to the Author):

I have no further concerns.

Reviewer #2 (Remarks to the Author):

With this re-revised version, the authors have further improved their manuscript significantly. Most importantly, the discussion and interpretation of results appears much more balanced to me. This is highly important and - I am sure - will trigger new research in the field. Great job!
No further comments from my side.

Reviewer #3 (Remarks to the Author):

I am happy that the authors have clarified several issues in the manuscript and collected new supporting data. They have also provided more moderation, especially in the discussion.